# Unsupervised classification of the northwestern European seas based on satellite altimetry data

Lea Poropat[1,2], Dan(i) Jones[3], Simon D.A. Thomas[3,4], and Céline Heuzé[1]

[1]Department of Earth Sciences, University of Gothenburg, Gothenburg, Sweden
[2]Now working at the National Centre for Climate Research, Danish Meteorological Institute, Copenhagen, Denmark
[3]British Antarctic Survey, NERC, UKRI, Cambridge, UK
[4]Department of Applied Mathematics and Theoretical Physics, University of Cambridge, Cambridge, UK

**Correspondence:** Lea Poropat (lep@dmi.dk)

**Abstract.** From generating metrics representative of a wide region to saving costs by reducing the density of an observational network, the reasons to split the ocean into distinct regions are many. Traditionally, this has been done somewhat arbitrarily, using the bathymetry and potentially some artificial latitude/longitude boundaries. We use an ensemble of Gaussian Mixture Models (GMM, unsupervised classification) to separate the complex northwestern European coastal region into classes based on sea level variability observed by satellite altimetry. To reduce the dimensionality of the data, we perform a principal component analysis on 27 years of observations and use the spatial components as input for the GMM. The number of classes or mixture components is determined by locating the maximum of the silhouette score and by testing several models. We use an ensemble approach to increase the robustness of the classification and to allow the separation into more regions than a single GMM can achieve. We also vary the number of empirical orthogonal function maps (EOFs) and show that more EOFs result in a more detailed classification. With three EOFs, the area is classified into four distinct regions delimited mainly by bathymetry. Adding more EOFs results in further subdivisions that resemble oceanic fronts. To achieve a more detailed separation, we use a model focused on smaller regions, specifically the Baltic Sea, North Sea, and the Norwegian Sea.

## 1 Introduction

Sea level variability in coastal regions is a critical area of research due to its implications for coastal management, climate change assessments, and hazard mitigation (Fox-Kemper et al., 2021). Sea level also reflects ocean currents (Dangendorf et al., 2021), so understanding the patterns and classifying the ocean based on sea level data can provide valuable insights into the dynamic behavior of these regions. Furthermore, while satellite-based instruments provide us with observations covering large areas, they only exist for the last three decades at most (Ablain et al., 2015), thus in order to study interannual and decadal processes, we have to rely on tide gauges, which are only available at specific point locations. Knowing the regions of coherent sea level variability allows us to estimate how broad of an area our conclusions based on tide gauges can be applied to.

The traditional ways for studying coherence in sea level is by calculating correlation, a principal component analysis (PCA) or a combination of the two. For example, Papadopoulos and Tsimplis (2006) extracted empirical orthogonal functions (EOFs) to create regional indices that represent sea levels in large areas of the world oceans and calculated the correlations with

climate indices to study the teleconnection patterns. Bulczak et al. (2015) used EOFs to decompose the observed seasonal
sea level variability in the Nordic Seas and compare it with the steric and dynamic forcing. Iglesias et al. (2017) performed a
correlation analysis between the altimetry-observed sea level anomaly in the North Atlantic and the teleconnection patterns.
As most studies do, they all separated the ocean into regions only based on geographical locations, i.e., the coastlines and
ocean basins, but did not attempt to further separate the basins based on the differences in observed sea level variability. There
have also been a few attempts to apply more complex classification or clustering methods to sea level, e.g. Scotto et al. (2010)
used agglomerative hierarchical methods to group time series in the North Atlantic Ocean based on their posterior predictive
distributions for extreme values, Thompson and Merrifield (2014) applied it to the whole ocean, while Barbosa et al. (2016)
used wavelet-based clustering to find regions with similar sea level records in the Baltic Sea. Self-organizing maps (SOMs;
Kohonen and Mäkisara, 1989; Kohonen, 1990) are a type of unsupervised neural network often used for clustering and pattern
analysis in atmosphere and ocean research. They have, among many other things, been successfully applied to find the patterns
of upper layer ocean circulation from altimeter observations on the West Florida Shelf (Liu and Weisberg, 2005) and in the
South China Sea (Liu et al., 2008), as well as from radar data in the northern Adriatic (Mihanović et al., 2011). Camargo et al.
(2023) applied SOM, as well as a network detection approach ($\delta$-MAPS) to regionalize the world's sea level budget. However,
SOMs are primarily a feature detection tool, which is also able to perform classification. Since SOMs are based on a neural
network, it is harder to interpret the results.

Therefore, in this work we use another unsupervised classification method called the Gaussian Mixture Model (GMM;
Bilmes, 1998) to determine the regions of coherent sea level variability. This method has already been used in oceanography
to classify the ocean based on temperature and salinity profiles. Maze et al. (2017) applied it to temperature profiles in the
North Atlantic to find the regions with similar vertical thermal structure, and Jones et al. (2019) did a similar study of the
Southern Ocean. Rosso et al. (2020) focused only on a part of the Southern Ocean, the Kerguelen Sector, but included both
the temperature and the salinity observations into the model. Thomas et al. (2021) then did a similar study of the whole
Southern Ocean. They all used PCA to reduce the number of levels in the vertical, which reduces the computational cost
of the classification. Here we apply the same method on satellite observed sea level, using PCA to reduce the amount of
information in the temporal domain. GMM provides similar output as the self-organizing map, i.e., the classification of the
area and the main pattern for each class, but since it is based on statistical distributions, it is easier to interpret the results.
Because clustering methods such as GMM give a class for every data point, the results from it not only provide an insight into
the patterns of sea level variability, but can also be used as a mask to isolate a region and focus on the dominant processes in
it without being affected by the noise from everything in the neighboring areas. GMM is also probabilistic, i.e., it provides the
probability distribution across all classes for each data point, which can be helpful when trying to determine the robustness of
the classification, giving it an advantage over simpler approaches, such as K-means clustering (Lloyd, 1957).

In Sect. 2, we describe the methods and data used in this paper. We start with the description of the used data set and
applied data processing steps (Sect. 2.1), then we continue to explain how the Gaussian Mixture Model works (Sect. 2.2), and
finally detail the ensemble classification procedure (Sect. 2.3). Sect. 3 contains the results and the discussion, focusing on the
classification and its dependence on the amount of information contained in the data set (Sect. 3.1), showing the results for

specific subregions of our area of interest (Sect. 3.2), and then illustrating how the classification works in the abstract empirical orthogonal function domain in which it is performed (Sect. 3.3). Finally, we give a summary of our work and present the conclusions in Sect. 4.

## 2 Method

To determine the regions of coherent sea level variability we use a machine learning method called Gaussian Mixture Model (GMM). It is an unsupervised classification (or clustering) model, i.e., a model that seeks to sort data points into classes due to their similarity without any *a priori* information about the classes. We apply it to the satellite-observed sea level data in the northwestern European coastal seas and part of the Atlantic Ocean. To increase the robustness of the classification, we use an ensemble of GMMs.

### 2.1 Data preparation

For the sea level variability information we use gridded reprocessed global ocean sea surface height satellite observations downloaded from Copernicus Marine Services (Pujol and Mertz, 2020). This data set uses a multi-mission mapping procedure based on an optimal interpolation technique derived from Le Traon and Ogor (1998), Ducet et al. (2000), and Traon et al. (2003), which combines the data from all available satellite missions: Sentinel-3A/B, Jason-3, HY-2A, Saral[-DP]/AltiKa, Cryosat-2, OSTM/Jason-2, Jason-1, Topex/Poseidon, Envisat, GFO, ERS-1/2 (Taburet et al., 2019). The data set has a global coverage, with 0.25° spatial and monthly temporal resolutions. The full description of the processing of the altimetry data and all the corrections applied to them can be found in Pujol et al. (2016) and Taburet et al. (2019).

As an example for our method, we select the area between 10°W and 30°E, and 50°N and 75°N (Fig. 1), which covers the coastal seas of northwestern Europe and a part of the North Atlantic Ocean. It is an interesting and complex region that comprises of many different ocean floor features and includes both mid-latitudes and polar region, as well as continental shelves and deep ocean regions. It consists of the very shallow enclosed Baltic Sea, the shallow North Sea and coastal seas between Great Britain and Ireland, the Faroe Shelf, the Norwegian continental shelf and part of the Barents Sea, all shallower than 1000 m. At the other extreme in depth, this region of interest also includes the Norwegian Sea, part of the Greenland Sea, and a section of the mid-Atlantic ridge in between, all deeper than 2000 m.

We use 27 years of data, from 1995 to 2021, to avoid the large areas with missing data in year 1994. In a few winters there are small gaps in the data in the Gulf of Bothnia and Gulf of Finland in the Baltic Sea, due to the extensive sea ice cover, which prevents sea level retrieval through altimetry (Pujol et al., 2016; Taburet et al., 2019). We linearly interpolate these grid points in time as the first step of data processing, in order to use the whole available area. We also remove the seasonal cycle by subtracting the climatology calculated from the whole 27 years long time period in order to focus on the non-seasonal variability. The data set still contains the trend, i.e., the sea level rise signal and its spatial patterns.

Although it is technically possible to use the time series directly as input for the mixture model, as it is with SOMs (Liu and Weisberg, 2005), they contain so much noise that the model is unable to converge to one best distribution of classes. It also

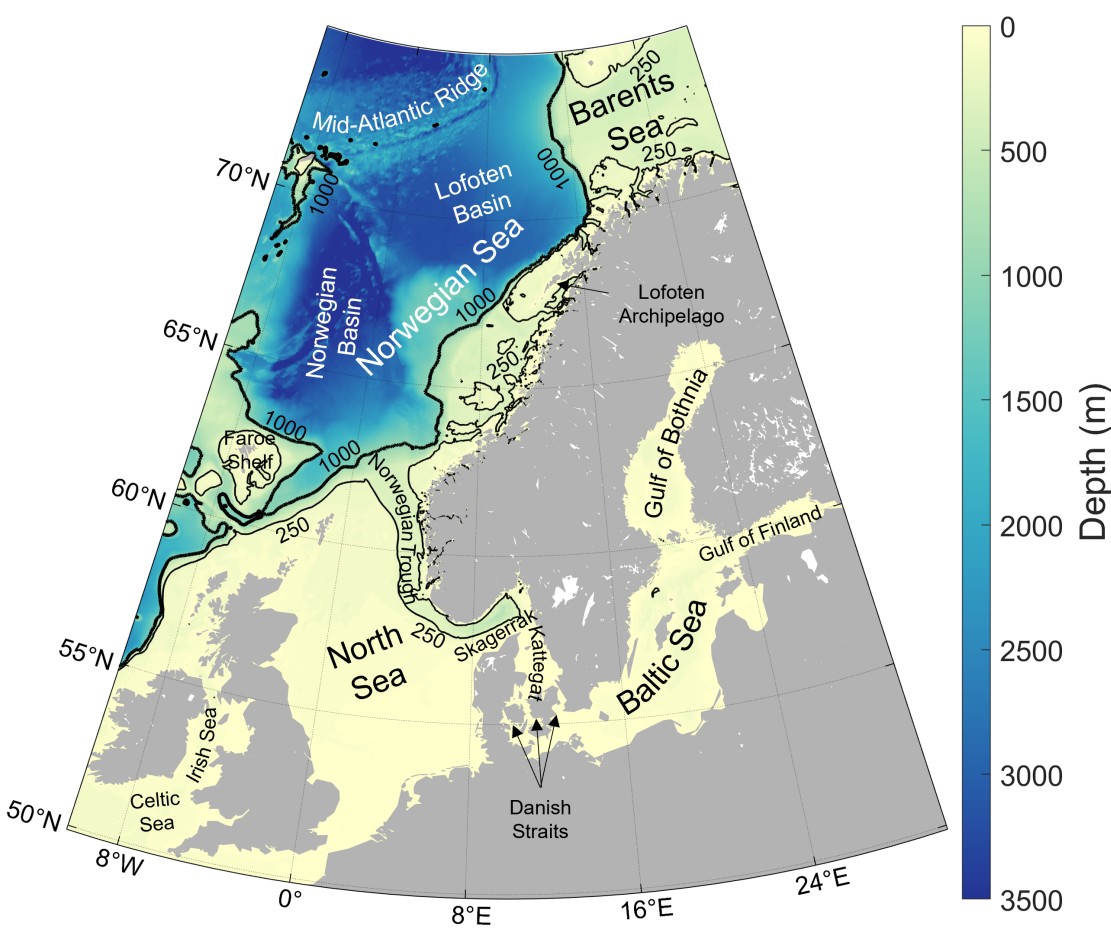

**Figure 1.** Our region of interest and its bathymetry, from the General Bathymetric Chart of the Ocean GEBCO 2022, along with place names referred to in the manuscript. Black contours represent the 250 (thin line) and 1000 m (thick line) isobaths.

makes the model one order of magnitude slower, from approximately 1 to 13 seconds for a single GMM, which makes testing large ensembles much more time consuming. Therefore, before applying the unsupervised classification method, we perform a principal component analysis (PCA) on the altimetry data set to reduce its dimensionality. This is a standard procedure when applying GMM to other data sets (e.g., Maze et al., 2017; Thomas et al., 2021), but is rather uncommon when using other clustering techniques on sea level data, where previous studies typically used the whole time series (e.g., Liu and Weisberg, 2005; Liu et al., 2008). We obtain the empirical orthogonal function (EOF) maps, which contain the spatial component of the dataset, and the accompanying principal component time series, as described in Björnsson and Venegas (1997), and use the

EOF maps as input for the machine learning classification model. In this way the input data is reduced from 324 monthly grids to only 3-9 EOFs, which explain approximately 75-85% of observed variability, and each grid point that we wish to classify is explained with 3-9 values instead of by the whole 27 years long time series. The decision on how many EOFs are included is based on whether we are trying to achieve a simpler classification, in which case less information is enough, or study the finer details, which requires higher degree EOFs. We train all our models on 90 % randomly selected grid points and use the remaining 10 % as a test set, to ensure that the model is not only fitted to the training set but is able to generalize to the data points that were not used for training.

## 2.2 Gaussian mixture model

We want to objectively identify patterns appearing in the EOF maps of the satellite-observed sea level and use them to define regions of similar sea level variability. A powerful method for this task is the Gaussian Mixture Model, an unsupervised machine learning classification approach that provides the probability that a location belongs to each of the classes. Since it is an unsupervised method, it does not need any prior information about the classes. It is based solely on sea level variability, without any geographical information, which allows us to determine physically coherent regions even if they are not adjacent. Finally, since this method provides the probability distribution across all of the classes, it enables us to distinguish clearly-coherent regions from boundaries.

GMM is based on the assumption that any probability density function (PDF) can be described with a model of weighted sums of Gaussian PDFs, which represent the components of the mixture model. In our case, the PDF describing the sea level EOFs can be represented with a weighted sum of Gaussian PDFs:

$$p(\boldsymbol{x}) = \sum_{k=1}^{K} \lambda_k \mathcal{N}\left(\boldsymbol{x}; \boldsymbol{\mu}_k, \boldsymbol{\Sigma}_k\right), \tag{1}$$

with $K$ components, where

$$\mathcal{N}\left(\boldsymbol{x}; \boldsymbol{\mu}_k, \boldsymbol{\Sigma}_k\right) = \frac{1}{\sqrt{(2\pi)^D |\boldsymbol{\Sigma}_k|}} \exp\left(-\frac{1}{2}(\boldsymbol{x} - \boldsymbol{\mu}_k)^\top \boldsymbol{\Sigma}_k^{-1} (\boldsymbol{x} - \boldsymbol{\mu}_k)\right) \tag{2}$$

is the multivariate Gaussian distribution in $D$ dimensions with mean $\boldsymbol{\mu}_k$ and covariance matrix $\boldsymbol{\Sigma}_k$. The weighting coefficients $\lambda_k$ must satisfy $0 \leq \lambda_k \leq 1$ and $\sum_k \lambda_k = 1$ (Bishop, 2006). The classification occurs in the abstract, $D$-dimensional EOF space, where each dimension represents one of the EOFs and each data point $\boldsymbol{x}$ is a grid cell described by $D$ EOF values. The aim of the GMM is to fit the PDF model from Eq. (1) to the observed probability density function of the EOFs by maximizing the likelihood of the observations using the Expectation-Maximization method (Dempster et al., 1977; Bishop, 2006), which is referred as model training. This boils down to finding the best estimates for the parameters $\lambda_k$, $\boldsymbol{\mu}_k$, and $\boldsymbol{\Sigma}_k$. After the observed PDF has been decomposed into a sum of $K$ Gaussian mixture model densities defined by mean $\boldsymbol{\mu}_k$ and covariance matrix $\boldsymbol{\Sigma}_k$ with $\lambda_k = p(c = k)$ being the component *a priori* density for class $c$, we then use Bayes' theorem:

$$p(c = k | \boldsymbol{x}) = \frac{p(\boldsymbol{x} | c = k) p(c = k)}{p(\boldsymbol{x})}, \tag{3}$$

where $p(\boldsymbol{x}|c = k)$ is defined by Eq. (2) and $p(\boldsymbol{x})$ is given in Eq. (1), to obtain the *a posteriori* probability of a location belonging to class $c$:

$$p(c = k|\boldsymbol{x}) = \frac{\lambda_k \mathcal{N}(\boldsymbol{x}; \boldsymbol{\mu}_k, \boldsymbol{\Sigma}_k)}{\sum_{j=1}^{K} \lambda_j \mathcal{N}(\boldsymbol{x}; \boldsymbol{\mu}_j, \boldsymbol{\Sigma}_j)}. \tag{4}$$

Finally, the location is labeled with class $k$ for which the posterior probability is the largest. The mean values which define each class in our case give us information about which EOFs and, consequently, if we are able to identify which processes a particular EOF represents, which processes are dominant in that region. A detailed description of Gaussian Mixture models can be found in e.g., Bilmes (1998) or Bishop (2006), while e.g., Maze et al. (2017) or Thomas et al. (2021) provide similar, oceanography oriented, explanations. The computation is performed using the open-source Python library Scikit-learn (Pedregosa et al., 2011).

In addition to deciding on the amount of information included in the data by selecting the number of EOFs $D$, the only input parameter to the GMM is the number of mixture components or classes, $K$, which needs to be specified before applying the model. It is not an easy task to determine the appropriate $K$. Sometimes it is possible to select a number by relying on theory behind the processes we are studying, e.g., when using GMM to find the fronts in the Antarctic Circumpolar Current like in Thomas et al. (2021), the number of fronts is known, but that is not always possible. There are multiple methods to objectively determine the optimal $K$, of which we use the silhouette score (Rousseeuw, 1987). Silhouette score for each sample (grid point) is computed as:

$$S_i = \frac{b - a}{\max(a, b)}, \tag{5}$$

where $a$ is the mean intra-cluster distance between the sample $i$ and all other samples from the same cluster and $b$ is the mean nearest-cluster distance between sample $i$ and all samples from the nearest cluster. To determine the best number of classes, we use the $S$ averaged over all samples. $S$ ranges between $-1$ and 1, where higher values correspond to better distinguished classes. Other studies, such as Maze et al. (2017) or Thomas et al. (2021), used the Bayesian Information Criterion (BIC) for this purpose. In our case however, the $K$ selected based on the BIC is usually too large (see next subsection), while the silhouette score provides a better estimate. However, since the silhouette score does not always work perfectly, we test all the class numbers between 2 and 11 for each number of EOFs and our tests confirm that the silhouette score in our case indeed recommends the best option. The summary of the tests is given in Table A1.

## 2.3 Ensemble classification

The initial class means in the GMM algorithm are determined by the simpler k-means clustering method, which depends on random initialization. To test whether the model converges, we do not specify random seed, so the initial parameters are different every time. Due to the size and complexity of the area and the sea level variability, each time the model is trained the results can be slightly different. Despite sometimes resulting in different classifications, the probability given by the GMM is almost always very close to one inside the class and lower only along the class borders, making it hard to assess which classification is better. To mitigate that and increase the robustness of the results, we use an ensemble prediction. Since GMM

provides the probability for a point to belong to each of the classes, the most fitting way to do the ensemble classification is
to use soft voting (Cao et al., 2015). With this method, the ensemble takes into account the probabilities from each model that
a point belongs to each class, the class with the largest sum of probabilities wins and the grid point is finally assigned to that
class. We also obtain the likelihood that a grid point will belong to that class, i.e., a combination of the number of models that
sorted it into that class and the probability they provided, which tells us how difficult it was for the model to sort that particular
location.

For most unsupervised classification models, including GMM, the main problem with using an ensemble is that, since
the classes are not known *a priori*, they are not numbered in any particular way, so class 1 of one ensemble member can
correspond to class 7 of another. Considering there are some differences between model runs, it is also possible that a class
appears in some model runs but not others. To be able to compare the classes from all ensemble members, we match them
based on the correlation between the class means (a $D$-dimensional vector), which results in a list of classes that appeared at
least once in any of the models of the ensemble. This list of classes is substantially longer than the predetermined number of
components $K$, but after voting, many of the classes get voted out because they only appear in a few of the models. Please note
that matching classes based on correlation requires at least three points; using the ensemble in this manner does not work for
only one or two EOFs.

Another problem that appears when using an ensemble is that, while GMMs themselves usually do not result in very small
classes, after soft voting, some classes can lose most of their points to neighboring classes, ending up with only a few data
points. That is avoided by setting up a minimal class size threshold, excluding those classes that have a number of points below
the threshold, and re-sorting all the grid points belonging to the excluded classes to the class with next highest vote. In such
cases, there are usually two classes with very similar probability sums, so the resulting likelihood is not considerably reduced.

Therefore, there are three parameters pertaining to the ensemble: the minimal correlation for the class means to be considered
"same", the minimal class size, and the number of ensemble members $N$. The other parameters that we need to set are the
GMM's only intrinsic parameter; the number of classes $K$; and the amount of information included into the model set by the
number of EOFs $D$. To determine which combination works best, we use three criteria: (1) the model converges, i.e., multiple
experiments with the same parameters find the same classes; (2) the ensemble keeps the same number of classes prescribed to
the individual GMM; and (3) the average likelihood inside the classes is as high as possible for the desired level of subdivision,
while still allowing low likelihoods on class borders. After testing several options, we find that the ensemble works best if we
use a minimal correlation for matching classes of $0.98$. With smaller correlation, classes that are not similar enough could be
merged, while with larger, even a slight difference in geographical distribution of a class in different ensemble members results
in the ensemble seeing them as different classes, neither of which receives enough votes, so only the next best class wins. The
minimal class size should be chosen based on the smallest area we are trying to capture, so we select a size of 100 grid points,
which allows the model to sort the small basins such as the Gulf of Bothnia or channels such as the Kattegat into a separate
class if necessary. We use 200 ensemble members for all our experiments. In some cases with small number of classes, we
would have achieved the same results with less, but since training a single GMM is fast, using an ensemble with 200 members
does not take too much time and it increases the robustness of the results. Using more ensemble members usually does not

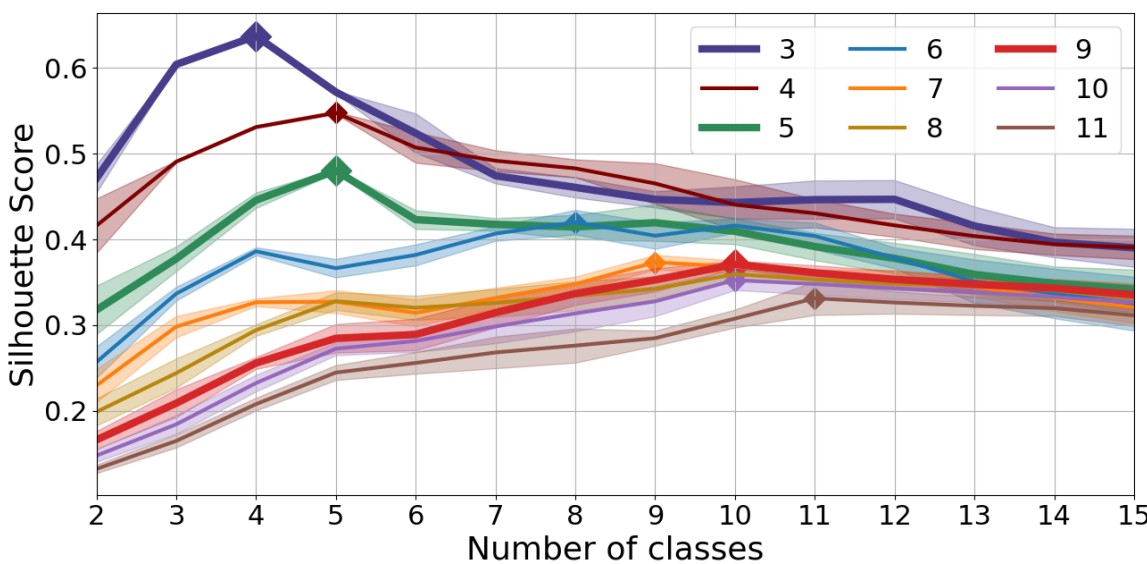

**Figure 2.** Silhouette Score for different numbers of classes calculated for Gaussian Mixture Models using different numbers of empirical orthogonal functions (differently colored lines). The class for which a respective model has the highest silhouette score is marked with a diamond. Silhouette score is computed for 100 models using the same parameters and the figure represents their average values (central lines) and one standard deviation (shaded areas). The models are fitted to randomly selected 90% of the grid points from the region shown in Fig. 1. Thicker lines and larger markers represent the models presented in this paper.

improve the results, with the exception of 11 EOFs and 11 classes (Table A1). The randomly selected 90 % of the grid points used for training are the same for each ensemble member, leaving the completely independent 10 % of the data for testing of the ensemble.

In the end, we obtain an ensemble classification with a new class number $K_E$ that is usually similar to, but not necessarily exactly the same as, the *a priori* class number $K$, along with a likelihood that a particular point belongs to the selected class.
The likelihood would correspond to 1 if all ensemble members chose that particular class with the probability of 1. Both fewer models assigning that class and models assigning it with a lower probability, i.e., the models not being certain that the grid point belongs there, reduce the likelihood. This is a big advantage compared to using only individual GMMs because it makes it easier to see how stable each classification is.

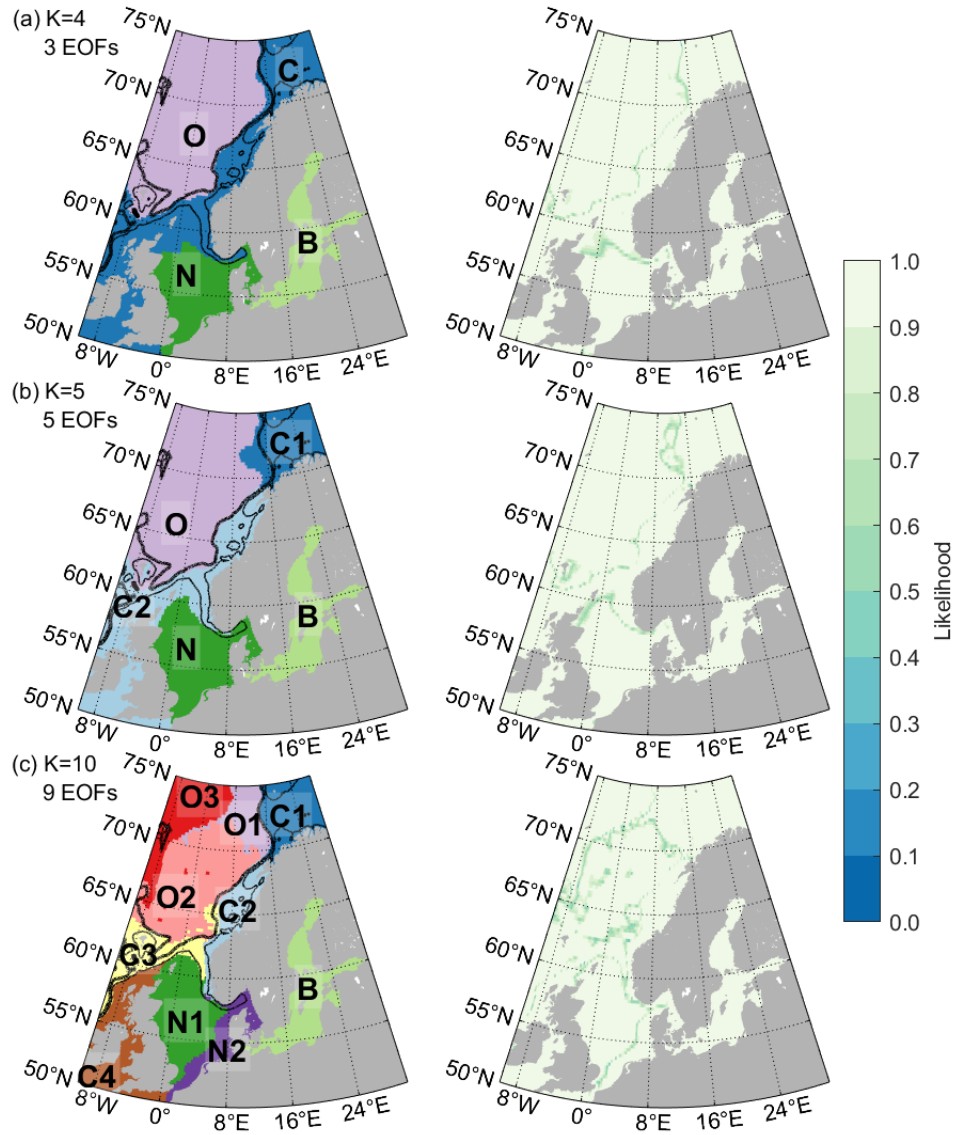

**Figure 3.** Classification using an ensemble of 200 Gaussian Mixture Models (left) and the respective likelihoods of the model sorting the grid points to that particular class (right). Classification is performed using 3 (a), 5 (b), and 9 (c) empirical orthogonal functions and 4, 5, and 10 classes, respectively. Letters indicate the names used to refer to the regions in the core of the text. Contour lines represent the 250 and 1000 m isobaths.

## 3 Results

### 3.1 Classification depending on the number of empirical orthogonal functions

After greatly reducing the dimensionality of the data with principal component analysis, the question arises as to how much data we should keep. As can be seen in Fig. 2, the optimal number of classes based on the silhouette score grows as we add more information, i.e., more EOF maps (colors), to the mixture. We can therefore decide on both (1) the number of EOFs and (2) the number of classes based on how many details we need to retain for our application. As we increase the number of dimensions (EOFs), the distance between any two points becomes more similar and less meaningful, which also lowers the silhouette score for all class numbers for a given high number of EOFs, so it does not make sense to compare the silhouette score for different number of EOFs. Note that according to the silhouette score, sometimes adding another principal component does not increase the number of classes the model is able to support, but it could still change which classes the model decides to include with this newly added information. Testing the models confirms that the number of classes recommended by the silhouette score is correct, as can be seen from Table A1 in the appendix. Typically, when using high number of EOFs with a smaller number of classes or vice versa, the models either do not work at all, i.e., re-running the ensembles results in a different classification, or work, but have lower likelihood than the models we selected. In some cases when the difference between the number of classes $K$ given to the individual GMMs and the optimal number of classes is small, the ensemble is able to find the optimal number of classes on its own.

Fig. 3 shows the classification obtained using 3 (simplest model), 5 (intermediate), and 9 (complex) EOF maps, which contain 76%, 80%, and 84% of the observed variability, with an ensemble of 200 GMMs. All classifications are created using the number of classes recommended by the silhouette score: 4, 5, and 10 respectively, which are indeed the numbers with which the model works best. Since the ensemble classification is able to modify the number of classes if the chosen one does not work well by discarding the classes that are only rarely selected by individual models, the fact that the ensemble maintains the selected number of classes is an additional proof that the number is good. It can be seen that even though the model does not consider the geographical information at all, it properly sorts all the grid points, both from the training set and the test set, into suitable geographically connected areas.

In the simplest model based on only three EOF maps (Fig. 3a), the GMM splits the area into only four classes: Baltic Sea (B), North Sea (N), which includes most of the transition area towards the Baltic called Skagerrak and Kattegat, the remaining continental shelf areas, including the northernmost part of the North Sea and the shallow Barents Sea (C), and the deep open ocean (O). The border between the latter two classes follows almost perfectly the continental shelf border, which can be seen from the 1000 m isobath. The border between the Baltic and the North Sea classes is also related to the geographical properties and is set at the narrowest region connecting them, the Danish Straits. Note that despite these class borders coincide perfectly with the steep changes in the ocean depth or with the coastlines, the GMMs do not explicitly include those things; the classification is based solely on the differences in sea level variability caused by different dominant processes on the continental shelf and in deep waters, as well as by the coastlines directing the circulation in the enclosed seas. The steric contribution is

known to be prevalent in the deep ocean, while in coastal regions complex bathymetry, local circulation, and forcing from the atmosphere and rivers can be more significant (e.g., Passaro et al., 2015).

The only border that does not seem to be directly caused by bathymetry is between the majority of the North Sea (N) and the remaining coast (C). That border is also the hardest one to classify, which can be seen from the likelihood (as low as 0.29); it is the only area where a significant number of models creates a slightly different border between the classes. It is most likely related to some of the underlying mechanisms in that region, such as the poleward propagation of sea level fluctuations along the eastern boundary of the North Atlantic, as found by Chafik et al. (2023) or the variations in the Atlantic inflow into the North Sea (Winther and Johannessen, 2006). North Sea sea level is also highly affected by wind and atmospheric pressure, and which of them dominates depends on the location (Dangendorf et al., 2014). The border corresponds well with the border found by Mangini et al. (2021) between the dominant influence of different jet clusters which represent large-scale atmospheric circulation patterns. Despite having one border which is harder to define for individual GMMs, the ensemble classification is extremely robust; increasing the number of classes in the individual ensembles to five or even six results in the exact same classification because the ensemble removes the unnecessary classes. This shows that based on the largest processes contained in the first three EOFs, there are exactly four regions with distinct sea level patterns in the northwestern European seas.

Adding two more EOFs that together contribute 4 % of variability information (intermediate model, Fig. 3b) does not change the classification significantly. The class borders from the simplest classification remain principally the same, and the new EOFs only allow the separation of the Barents Sea from the remainder of the coastal class. The border between the North Sea (N) and the adjacent coastal class (C2) is also moved slightly northward.

Finally, when using 9 EOF maps (the most complex model, Fig. 3c), we end up with 10 classes in our region. The class borders due to bathymetry or the processes related to it remain the same. There is further subdivision of both coastal and open ocean areas, and the border of the North Sea -itself subdivided, is shifted even further northward and now coincides with the 250 m isobath which in that region marks the edge of the Norwegian Trench. Models maintaining the basic classification and further subdividing some of the classes after adding new information into the model is not a characteristic of the GMM. The GMM could completely change some or all the classes if the number of classes is different, so the fact that this is not happening here must be based on the characteristics of the ocean. The ocean is first coarsely divided into regions determined by the bathymetry, and then each of those regions can be further subdivided based on other aspects of the sea level variability such as finer resolution bathymetric features (e.g. the subdivision of class O around the mid-Atlantic ridge) or water masses (e.g. the Barents Sea separating from class C).

The virtually same classification as in Fig. 3c can likewise be obtained with 8, 10 or 11 EOF maps. With 11 EOFs and a significantly larger ensemble of 1000 members it is also possible to achieve a separation into 11 classes (not shown), in which the Barents Sea (class C1) is further split into northern and southern. Beyond that, adding even more EOFs does not result in a finer subdivision; it only introduces so much noise that multiple ensembles with the same parameters produce different results. With that in mind, if we would like to obtain a more detailed subdivision, it is better to do the classification for a smaller region. The complexity of sea level patterns differs significantly from region to region, so narrowing our focus to a smaller area would

allow us to use the principal components specific to that area, increasing the amount of information in fewer EOF maps, thus reducing the noise and allowing a better classification.

## 3.2 Reducing the size of the region

Here, we apply the ensemble GMM to three sub-regions of our area of interest: the Baltic, the North and the coastal part of the Norwegian Sea (Fig. 4). By using EOF maps calculated solely for these regions, the model input contains only the data relevant for them, without the noise coming from EOFs significant only elsewhere, which allows the models to find more region-specific patterns and increase the number of classes they are able to find. We select the models based on the principles presented in Sect. 2.3. We again calculate the silhouette score for all combinations of the number of EOFs and class numbers in all three sub-regions, but there are better class numbers than those recommended by the silhouette score (3, 5, and 7 classes). In the Baltic and the Norwegian Sea models the likelihood of the models we present is significantly higher than the likelihood of the model recommended by the silhouette score, while for the North Sea they are equally good, but the model we present has a higher number of classes, thus allowing a more detailed subdivision. This demonstrates that while the silhouette score is a good tool to give an estimate of the number of mixture components, it does not always give the best result. One should always try the model with several options to find the best solution to their specific classification problem.

The almost completely enclosed Baltic Sea is to a very large extent controlled by the variability at its entrance on a long-term scale (e.g., Lehmann et al., 2002), which is why it is also rather uniform compared to the other areas included into our whole region of interest and why the models based on the EOF maps from the whole area do not usually divide the Baltic Sea further than the basin-scale. This uniformity is also reflected in the EOFs calculated for the Baltic Sea separately: the first four explain 94.2% of the variability. The ensemble classification model based on them is able to distinguish two more classes (Fig. 4a) than in the most complex model based of EOF maps for the whole area of interest (Fig. 3c). The Baltic (class B in Fig. 3c) is now split into three classes: the Gulf of Bothnia (class 1), the western Baltic (class 3) and the remainder of the Baltic Sea (class 2). The Danish Straits, a series of narrow channels connecting the Baltic Sea with Kattegat, are now sorted together with the western Baltic class (class 3), while Kattegat and Skagerrak form one class connected to the North Sea (class 4). The likelihood is very close to 1 virtually everywhere except at some class borders, meaning that the majority of the ensemble members selected the same classes.

The North Sea has the most complex sea level variability patterns of the whole considered area. We would need almost 40 EOFs to explain the same level of variance as we do with only four for the Baltic. It is however enough to use only five of them, explaining 79 % of the sea level variability, to achieve a more detailed subdividion of the North Sea (Fig. 4b) than we can with the most complex model based on EOFs from the whole northwestern European coastal area (Fig. 3c). Class N1 from Fig. 3c is here further split into classes 2 and 3, and N2 into classes 4 and 5, of which 4 is the southern North Sea and 5 covers Kattegat and western Baltic Sea. The region-specific classification model also finds class 1, which mostly corresponds to class C3 from Fig. 3c, and it combines parts of classes C2 and C4 included here into class 6. The larger classes are separated as zones in the north-south direction, as expected from other works, e.g. Dangendorf et al. (2014) or Sterlini et al. (2016), who found a difference in the sea level variability between the northern and the southern North Sea. Some of the class borders (1

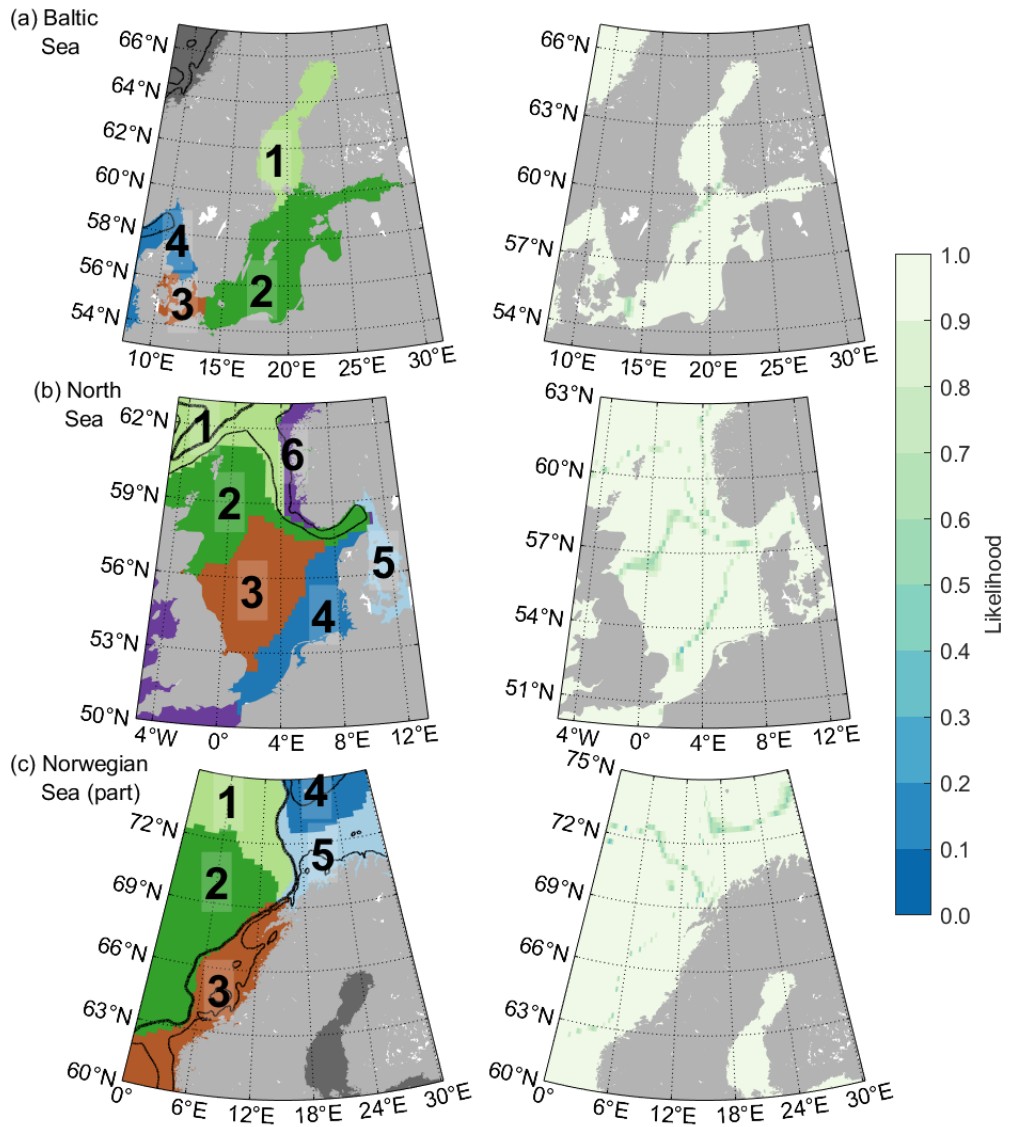

**Figure 4.** Classification using an ensemble of 200 Gaussian Mixture Models (left) and the respective likelihoods of the model sorting the grid points to that particular class (right) for the Baltic Sea performed using 4 EOFs and K = 5 (a); North Sea using 5 EOFs and K = 6 (b); and part of the Norwegian Sea using 6 EOFs and K = 6 (c). Numbers indicate the assigned classes. Contour lines represent the 250 and 1000 m isobaths. Dark gray classes in (a) and (c) have not been assigned a number because they are not part of the region of interest and are therefore not discussed in the text.

and 2) are also based on bathymetry, following the Norwegian Trough. Interestingly, part of the Norwegian coast included here is sorted into the same class as the western coast of Great Britain, which suggests that the model most likely sees some processes relevant for western coasts and the regional atmospheric pressure and wind patterns. The likelihood is close to one across the area, with the exception of class borders, which the ensemble members do not agree on so well. Some class borders from both the Baltic and the North Sea models match to a large extent with the classes obtained with the whole area model, in addition to which the region-specific models are able to further subdivide their regions. The classes in the overlapping area of the Baltic and the North Sea models, however, do not match, because the EOFs computed for the different regions do not capture the same processes.

Finally, the classification obtained by considering 6 EOF maps calculated for the Norwegian coast (Fig. 4c) is unfortunately unable to achieve a significantly more detailed classification than the most detailed model for the whole area (Fig. 3c). Classes 1, 2, and 3 from Fig. 4c generally correspond to classes O1, O2, and C2 from Fig. 3c, although there are some differences in class borders, particularly class 3 covers both the class C2 and part of C3 contained in the area of the Norwegian Sea model. The region-specific model also splits the Barents Sea opening based on its depth (classes 4 and 5), similar to the whole area model with 11 EOFs and 11 classes, which requires a much larger ensemble (not shown).

### 3.3 Empirical orthogonal functions

To learn more about how the GMM determines the classes, we can take a look at the empirical orthogonal functions (EOFs) because GMMs perform the classification based on them. Apart from assigning classes, GMM also gives the class means and covariance matrices it fits the data to, which in our case is a class mean for each EOF used to train it. Therefore, to see how the models from Fig. 3 determine the class borders, we can compare the EOF maps (Fig. 5a) with the maps in which we replace the values of EOFs at each grid point with mean values from the class assigned to that point (Fig. 5b, c, and d). This can reveal two things: 1) a comparison of the class mean EOFs with the original EOFs indicates how well the model fits to the data; and 2) the difference in class means between two classes can tell us which EOFs are responsible for that class border. The accompanying principal component time series can be seen in Fig. B1 in the appendix.

In the simplest model (Fig. 5d and Fig. 3a) the GMMs capture only the rough patterns of the first three EOF maps, which mostly represent the sea level rise (EOF1) and the North Atlantic Oscillation (EOF2 and EOF3). All three EOFs contain additional processes but those are not as easily identifiable. The border between the Baltic (class B) and the North Sea (class N) is visible on most EOF maps (column a), except the second one, showing that since the Baltic is an enclosed sea, almost all processes in it differ from those in the neighboring North Sea at least to some extent. The border based on the continental shelf break can also be seen on most EOFs, but is most visible in EOF 2 and 3, demonstrating how clearly the large difference in ocean depth affects sea level. The only border in the simplest model (column d) that is not based on bathymetry, i.e. the border between the North Sea and the rest of the continental shelf, just south of 60°N, is determined only by the not very steep gradients in the first three EOF maps in that location, which is probably why the individual GMMs do not completely agree where to place it, resulting in lower likelihood around the border.

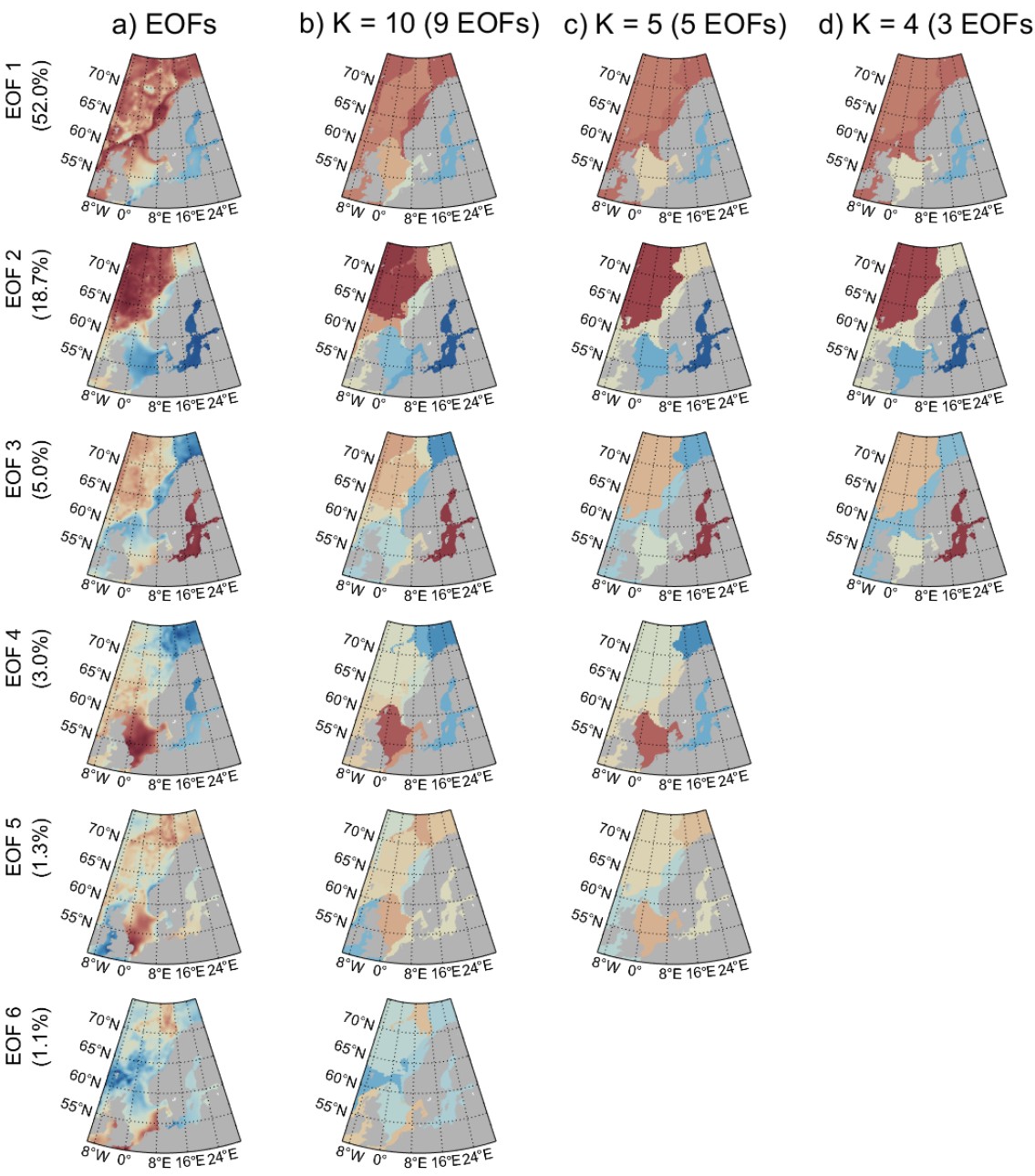

**Figure 5.** Empirical orthogonal function maps of satellite-observed sea level used as input for the Gaussian Mixture Model (a) and class means for the classification into 10 (b), 5 (c), and 4 (d) classes performed using 9, 5 and 3 empirical orthogonal functions, respectively. Column (b) contains only the first 6 EOFs of the 9 provided by the model, while columns (c) and (d) show the results for all EOFs used in the classification. Each row represents one of the EOFs and the color scale is the same for each plot, both the original and the three models, in that row.

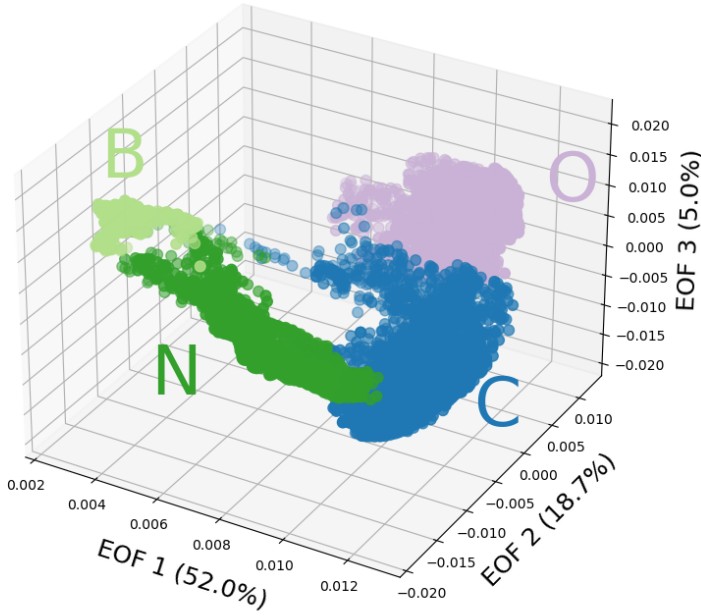

**Figure 6.** The classification analysis in the abstract EOF space for the model for the whole area with three EOFs and four classes shown in Fig. 3a. Each point represents a three-dimensional vector of EOFs that describe a single grid point and the three axes are the three EOFs. Class assignments are indicated using the same colors as in Fig. 3a.

When we look at the class means from the more complex models (Fig. 5, columns b and c), they start to resemble the original EOF maps more closely. Both the coastal area outside of the North and the Baltic Seas and the open ocean are rather uniform in the first three EOFs, which is why the simplest model is unable to divide them further. However, adding the fourth and fifth EOF map introduces a clear border in the continental shelf between the Barents Sea (C1) and the rest of the coastal shelf (C2). EOF 4, which has the most prominent signal in the North Sea, is responsible for shifting the northern border of the North Sea class northwards compared to the border in the simplest model. We need to add EOF 6, which is responsible for only 1.1 % of overall variability, to allow the most complex model to separate the southern part of the North Sea. This EOF mainly contains variability at periods of around 3 and 5-6.5 years, and has a very strong signal in the southern North Sea, but much weaker or negative everywhere else. It might in part represent the sea level variability related to the northern jet cluster found by Mangini et al. (2021), which dominates in the southern North Sea. The same EOF is also responsible for the separation of class O1 from the other deep ocean classes. The remaining separation of both deep ocean and coastal areas is done based on even higher level EOFs (not shown). Moreover, we can see that for the higher order EOFs, the class means become smoother, indicating that the

350 model learns less and less from each new EOF added to it, until it reaches the point when adding new EOFs introduces only noise and prevents the model from finding reasonable classes.

To achieve useful results, we need to find a balance between interpretability and accuracy. Simpler models with fewer EOFs tend to be easier to interpret, in that they have clearer boundaries between regions, but they fail to capture the full variability of the data. More complex models capture more of the variability of the data, but they tend to be harder to interpret, in that they

feature more ambiguous boundaries between regions. The compromise between interpretability and accuracy is not universal and should be tailored to the application at hand. In our case, the balance is struck when the classification is able to highlight novel ideas about the spatial coherence of sea level variability in our study region. Unlike the simplest and even the intermediate model, in which the class borders mainly coincide with bathymetry and could be determined without any particular method by selecting depth ranges or, in case of the Baltic Sea, based on the coastline, the most complex model separates the ocean

into regions that are not so obvious. The difference between the east, west and south coast in the North Sea is also found by (Dangendorf et al., 2014) and Frederikse and Gerkema (2018) based on tide gauges, but our work shows that their findings for the British coast are most likely also valid for most of the North Sea, while those for the Norwegian coast apply only on a narrow stretch along the coast. Furthermore, while different drivers of sea level variability in the North Sea have been already studied by many (e.g., Dangendorf et al., 2014; Frederikse and Gerkema, 2018; Hermans et al., 2020), the other shelf areas

and especially of the adjoining deep ocean basins are less well understood, so our results can help us determine which regions should be studied separately. Ultimately, unsupervised classification methods can be useful as "hypothesis generation tools" (Kaiser et al., 2022).

Since the simplest model for the whole area (Fig. 3a) is based on only three EOFs, it can be directly depicted in the abstract EOF space to see how the classes are distributed (Fig. 6). We can see that in the abstract space the model has a generally clear

separation between the classes and our conclusions about which EOF is responsible for which class border from the beginning of this subsection are confirmed. The Baltic class (B), being very uniform in regards to sea level variability, has a narrow range in the EOF space. North Sea class (N) is also quite uniform in the second and third EOF, but has a wide range in the first one because there is a large spatial gradient in the sea level trend there. The coastal class (C) is the most widespread of them, covering different levels of variability in the first three EOFs along the whole coastline. Finally, the open ocean class (O) covers

the largest area in space, but since it has less variability than the coastal class in the first three EOFs, it is less widespread in the EOF space. Even though the border between classes C and O is very compact in the EOF space, it is also apparently easily seen by the model and, when transferred to a map, almost perfectly follows the continental shelf break. This shows that, while some changes in variability might be small at the shelf break, they are very clearly defined.

## 4 Summary and conclusions

Gaussian Mixture Modeling, an unsupervised classification approach based on the assumption that all probability density functions can be described as a weighted sum of Gaussian PDFs, can be used to find regions of coherent sea level variability based on satellite altimetry data. Here, we focus on the northwestern European coastal shelf area and a small adjacent part of

the Atlantic Ocean, but the method is applicable in any region. While it is technically possible to use the time series of the sea level data directly as input for the GMM, that approach makes the fitting extremely slow and introduces too much noise for the model to converge towards a single classification solution. Using the empirical orthogonal function maps, the spatial part obtained with a principal component analysis, as input allows us to include most of the observed variability but with greatly reduced dimensionality and noise level.

After reducing the dimensionality, the GMM is able to separate our region of interest into a relatively small number of classes. However, if we want to use more than six mixture components (classes), the models start to diverge, with results varying slightly between individual model runs. Since the models generally find the same patterns despite some differences between them, we show here that we can use an ensemble approach to find the most common classification by applying soft voting, i.e., selecting the class which most models chose with a high probability. The ensemble also gives a likelihood of a model assigning this particular class for each grid point, which tells us how robust the classification is and how difficult it was for the models to classify each area. By comparing the class means with the EOF maps used as input for the GMM ensemble, we showed that we can usually see which class border is based on which EOF, making the model explainable to some extent, and thus directly useful for scientific analysis.

The simplest classification of our entire region of interest, i.e., the classification based on only 3 EOF maps, mostly follows the bathymetry and the coastlines. After including more EOF maps, this basic separation remains the same, but the models are able to also find class borders that are based on ocean dynamics. The largest number of classes with which we can achieve robust results for our region is 10, and we need to use between 8 and 11 EOF maps as input. This model finds two classes along the Norwegian coast, two in the North Sea, and only one for the whole Baltic Sea. Since the complexity of these three regions varies significantly between them, we show that we can achieve a much more detailed classification if we focus on each region separately.

We find that western Baltic has significantly different variability from the rest of the Baltic Sea and should be considered separately, as well as how to separate the North Sea from the rest of the continental shelf. We confirm previous findings that the North Sea has differences between eastern, southern, and western coast because of the different atmospheric drivers along those coasts, but also show that the sea level observed on the British coast is representative for the whole North Sea, while that observed on the eastern coast is more localized. We find that the sea level in the Barents Sea considerably differs from that along the rest of the continental shelf break but only after including higher order EOFs. We can use these results to further study these regions and determine what is the cause of different sea level changes in them.

This classification method is not based on any arbitrary threshold or even on the geographical information such as longitude and latitude, so it is applicable to other ocean regions. It could also be used for finding patterns of sea level variability on different temporal scales, both shorter, such as mesoscale eddies or storm surges, or longer, such as decadal changes or trends. It is not limited to altimetry observations; it could also easily be applied to *in situ* observations or to model data, to study past and future sea level variability that is changing in response to climate change. It can be used on its own, to gain more insight into the patterns of sea level variability, or just as a step in data processing, to create a mask for separating the ocean into

regions, which can then be further examined with other methods. Finally, the method is not limited to sea level, it could be used for any other variable.

*Code and data availability.* In this work we used satellite altimetry observations from Copernicus Marine Service (https://data.marine.copernicus.eu/product/SEALEVEL_GLO_PHY_L4_MY_008_047/description, downloaded on 2023-10-17). Bathymeric information is from the General Bathymetric Chart of the Ocean GEBCO (https://doi.org/10.5285/e0f0bb80-ab44-2739-e053-6c86abc0289c, accesed on 2022-11-07). Code for the ensemble classification with GMMs is available at https://zenodo.org/doi/10.5281/zenodo.10356063.

## Appendix A

**Table A1.** Average likelihood over the whole region from Fig. 1 for all the combinations of the number of empirical orthogonal functions (EOFs) and class numbers that produced stable results with an ensemble of 200 Gaussian Mixture Models. X marks the combinations for which the ensemble is unable to converge to the same classification result when testing it multiple times. The number of classes recommended by the silhouette score for each number of EOFs is indicted by bold numbers.

| No. of EOFs | Number of classes | | | | | | | | | |
|---|---|---|---|---|---|---|---|---|---|---|
| | 2 | 3 | 4 | 5 | 6 | 7 | 8 | 9 | 10 | 11 |
| 3 | 0.78 | 0.98 | **0.98**[1] | X | X | X | X | X | X | X |
| 4 | 0.44 | 0.98 | 0.98 | **0.97** | X | X | X | X | X | X |
| 5 | 0.39 | 0.63 | 0.92 | **0.98**[1] | X | 0.89 | X | X | X | X |
| 6 | X | 0.72 | 0.90 | X | X | 0.94 | **0.89** | X | 0.89 | X |
| 7 | X | 0.63 | 0.68 | X | X | X | X | **0.93** | X | X |
| 8 | 0.35 | 0.54 | 0.77 | 0.84 | X | X | X | X | **0.94** | X |
| 9 | X | X | X | 0.66 | X | X | X | X | **0.95**[1] | X |
| 10 | 0.31 | X | X | 0.70 | X | X | 0.70 | X | **0.96** | X |
| 11 | 0.36 | X | X | X | X | X | X | X | 0.94 | **0.96**[2] |

[1] Models shown on Fig. 3 and discussed in the manuscript.

[2] This ensemble required 1000 members to converge.

## Appendix B

*Author contributions.* LP and CH conceived the idea for this work. LP planned and performed the experiments, with the help of ST and DJ. LP, CH, and DJ analysed the results. LP wrote the manuscript, while all authors commented on it and provided critical feedback.

*Competing interests.* No competing interests are present.

*Acknowledgements.* This work is part of the research project "Would the Northern European Enclosure Dam really protect Sweden from sea level rise? (NEEDS)" funded by the Swedish Research Council for Sustainable Development (Formas, dnr 2020-00982, awarded to CH). LP also received funding from the Danish state through the National Centre for Climate Research (NCKF) in the revision stage of preparing this manuscript. DJ was supported by UK Research and Innovation (grant no. MR/T020822/1). ST was funded by UK Research and Innovation (grant no. EP/S022961/1). We thank the two anonymous reviewers for their comments that greatly improved the quality of the manuscript.

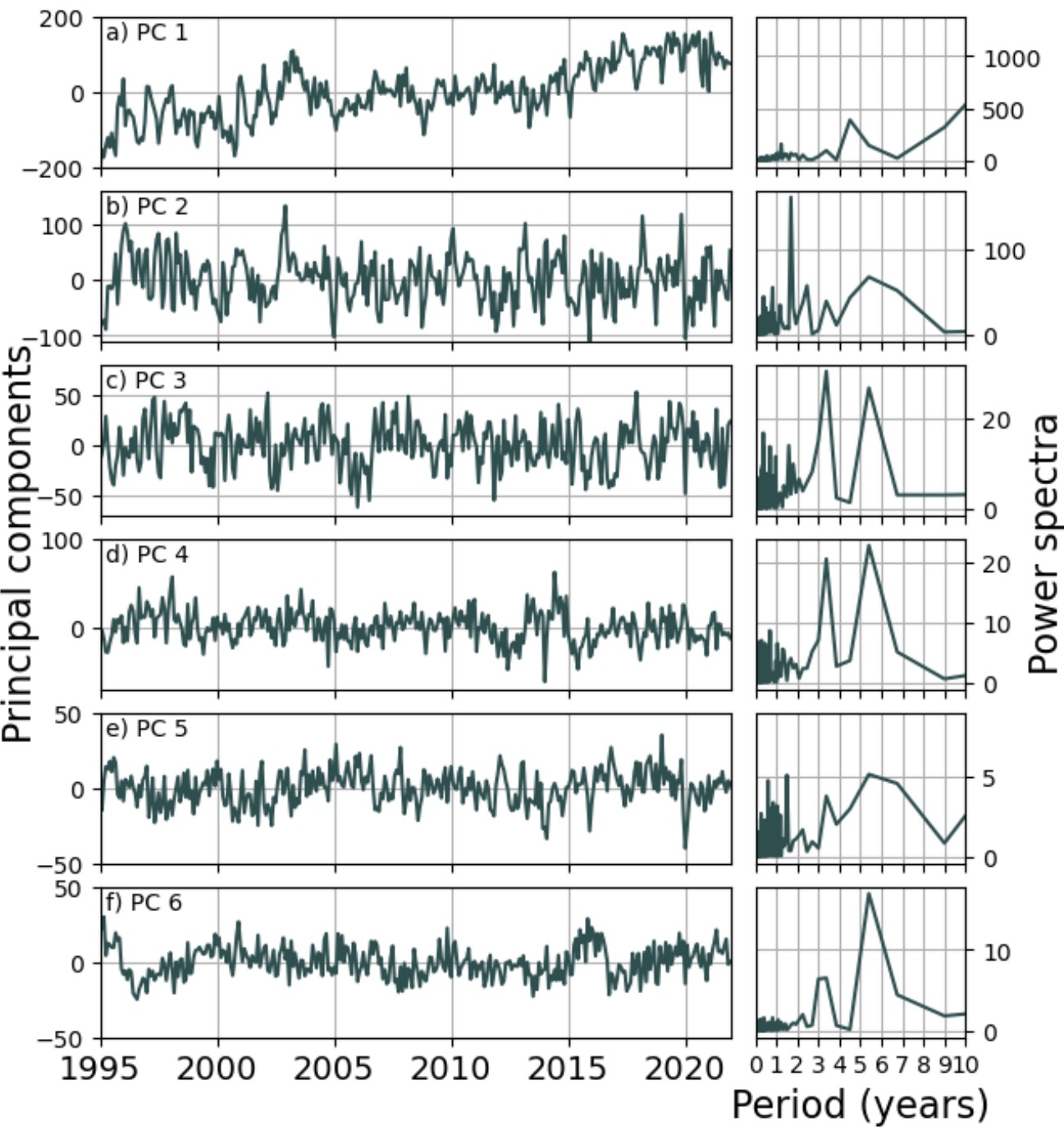

**Figure B1.** The first six principal component time series (left) and the accompanying spectra (right). They match the empirical orthogonal function maps shown in Fig. 5a.

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
