# Peer review of "Unsupervised classification of the northwestern European seas based on satellite altimetry data"

_EGUsphere, 2023_

## Author Comment (AC1)

**Response to Reviewer#1**

Review of "Unsupervised classification of the Northwestern European seas based on satellite altimetry data" by Poropat et al., 2023

In this work, Poropat and colleagues use a Gaussian Mixture Model (GMM) to identify coherent regions of sea-level variability in the Northwestern European Seas. They show how the number of EOFs and the number of classes (mixtures) in the models are important parameters that can result in different patterns, but the main classification remains the same, showing the robustness of their method. The work is focused on the method itself, and I personally missed a bit more discussion into what the identified patterns could actually mean. Nonetheless, I believe this is an important work and a good addition to the scientific community, which could be the base for more process-based studies in the future.

We thank the reviewer for their positive evaluation and valuable questions and comments that helped us to improve our manuscript. Note that the contribution of the reviewers has been duly acknowledged in the manuscript (see acknowledgement section). We added more discussion about the physical processes that might be related to the identified patterns. We address all the comments in the following text.

Before addressing the comments, we would like to mention that we changed the data set used in the whole manuscript in response to a question by Reviewer#2 about the time span of data we used to train our models. To address their comment, we downloaded the extended data set from years 1995-2021 and re-did all the experiments. However, in the time between we obtained the data used for the results presented in the first submission of the manuscript and now, the data set has also been re-processed (see description of the changes to the processing chain at https://catalogue.marine.copernicus.eu/documents/QUID/CMEMS-SL-QUID-008-032-068.pdf).
Both the different processing of the altimetry observations and to some extent the change in the time span used for model training affect the results of the classification. You can see the new results (replacement for Figs. 3 and 4 in the manuscript) in Figs. R1 and R2. In the cases where the reviewer's comment or our answer is affected by the change in the data set, we point that out in the reply to the specific comment.

**Summary of the changes in the classification results**

With the new data set, when considering the whole Northwestern European continental shelf (Fig. 3 manuscript and Fig. R1 here) the simplest model remains the same, the intermediate model has one fewer class (class O1 from the original manuscript is not visible in this model), and the complex model remains virtually the same. Class O1 is also found by this model, suggesting that the signals responsible for it are still present in the data, but now only in the higher EOFs, which is why a model using only five of them does not capture it. The area with low likelihood in the southern North Sea visible on Figure 3c in the original manuscript is also gone, suggesting that it was a result of data processing in the old data set. Additionally, with this data set, we obtain almost identical K=10 classification with 8, 9, 10, and 11 EOFs, of which we decided to display the results with 9 EOFs because it is the lowest number of EOFs (meaning the fastest model) for which the model reaches the minimal likelihood of 0.95 averaged over the whole region.

Of the sub-regions (Fig. 4 manuscript and Fig. R2) the Baltic Sea region classification remains the same and the classification of the North Sea has a difference only along the southern coast, most likely for the same reason as the difference in likelihood in the whole area model, but we now need 5 EOFs to achieve that instead of only 3. This unfortunately means that it is no longer possible to directly show this model in the abstract EOF space because it now requires 5 dimensions. Fig. 6 in

the manuscript, therefore, now has only the simple model of the whole region and the discussion of the classes in the abstract space is shortened.

[Figure]

**Figure R1:** Replacement for manuscript Fig. 3. Classification using an ensemble of 200 Gaussian Mixture Models (left) and the respective likelihoods of the model sorting the grid points to that particular class (right). Classification is performed using 3 (a), 5 (b), and 9 (c) empirical orthogonal functions and 4, 5, and 10 classes, respectively. Letters indicate the names used to refer to the regions in the core of the text. Contour lines represent the 250 and 1000 m isobaths.

[Figure]

**Figure R2:** Replacement for manuscript Fig. 4. Classification using an ensemble of 200 Gaussian Mixture Models (left) and the respective likelihoods of the model sorting the grid points to that particular class (right) for the Baltic Sea performed using 4 EOFs (a); North Sea using 5 EOFs (b); and part of the Norwegian Sea using 6 EOFs (c). Numbers indicate the assigned classes. Contour lines represent the 250 and 1000 m isobaths.

For the Norwegian Sea we did not manage to obtain the same number of classes as before, the highest K with stable results is 6. The likelihood is also lower than previously.

**Major comments**

> Number of EOFs X Number of Classes: The results from Section 3.1 are very interesting. But for me it wasn't clear if the different classifications are from adding more EOFs or from changing the number of classes. The authors used a "non subjective" way to choose the number of classes, which is important for several reasons.

> But I do wander, if the results from Figure 3 would be the similar if they fixed the number of EOFs, and just changed the number of classes. Was this tested? Because during the entire Section 3.1, the authors presents the results as an effect of adding more EOFs, but it could just be due to adding more classes. So testing the classification for fixed number of EOFs and changing the number of classes would make their results and discussion more robust.

We apologize this was not clearly explained in the paper, we added an explanation to the beginning of Sect. 3.1. We tested all class numbers from 2 to 11 for each of the presented numbers of EOFs (and for many not presented). The silhouette score is a very useful metric that often gives a good recommendation for the optimal number of classes, but it does not always work perfectly, so we tested all the combinations of number of EOFs and number of classes to confirm whether its recommendation was correct. In the end we picked the number of EOFs somewhat arbitrarily to show different levels of complexity and used the number of classes that work best for each given number of EOFs based on the criteria presented in Sect. 2.3., which in case of the whole Northwestern European shelf coincides with the recommendation of the silhouette score. In case of the sub-regions it does not, but the discussion about that is in the answer to one of your minor comments later in the text.

Normally, when using high number of EOFs with a smaller number of classes or vice versa, the models either do not work at all, i.e., re-running the ensembles results in a different classification, or work, but have lower likelihood than the models we selected. In some cases the ensemble also eliminates or creates a class to get to the best number, e.g., using K=4, 5 or 6 with 3 EOFs results in the same 4 classes because the ensemble eliminates the extra classes, which confirms that K=4 is indeed the best. We have included in the appendix a table with a summary of the results of all combinations of EOFs and Ks, with average likelihood over the whole region for all converging ensembles.

> Literature & Discussion: I missed the "discussion" section, but I don't think it's reasonable to ask the authors to add an entire discussion section, just maybe in some locations when describing the features identified, might be a good addition to refer to some papers that could bring some insights into the processes behind these patterns. For example, regarding the features with the classification, there are some works that could highlight some of the processes behind the identified patterns (e.g., Mangini et al, 2021; Hermans et al, 2020; Frederikse et al., 2018, Chafik et al (2023), Calafat et al (2013), among others). Also, I would expect the authors to acknowledge the works of Thompson & Merrifield (2014) and of Camargo et al (2023). Both works have performed classification of ocean regions based on sea level data, and seem relevant for the present work. The ocean regions from Thompson & Merrifield have been widely used in sea level studies. The work of Camargo et al (2023) used two classifying methods to identify coherent regions of sea level variability. One of the methods of Camargo et al (2023) was SOM, which Poropat et al mention on the introduction, and hence acknowledging this work there seems fitting.

Thank you for your comment and the references, they were very useful for improving the discussion of our results. We believe that an in-depth analysis of the mechanisms contributing to sea level variability requires applying additional methods and deserves a separate paper. GMMs on their own can only find patterns, not the causes of those patterns, so here we would like to focus on finding

regions of coherent sea level variability and in our next manuscript (currently under review) we focus on the driving processes of sea level in each of the regions we found here. We realize, however, that the manuscript should nonetheless include at least some discussion of the physical background and the mechanisms driving sea level, so we included more analysis of our results, trying to explain the dominant processes in at least some of the classes/regions and connect our findings with those from other works, including the references listed by the reviewer.

**Minor comments**

> L47-50: Isn't this true for other classification/clustering methods also? Once clusters are identified, it can be transformed in a mask to isolate regions…

Yes, it is true. We modified the sentence to make it clear that this is a possible use of all clustering methods, not specific to GMM.

> L93-94: It wasn't clear for me if it's common to use EOFs as input for GGMs, or if this was a "novel" approach that the authors found to reduce noise? Would be good to know in both cases.

It was used by all the studies we cited that were using GMMs with temperature and salinity profiles, so it is a common approach with GMMs. It is "novel" in regard to classification based on sea level; the studies we found that applied other clustering methods to sea level data usually considered whole time series. We rephrased the sentence to make this point clearer.

> L101-102: Just a comment, but this is also true for SOM.

True, they are similar methods in that regard.

> L124: How would the mean values give information about processes associated? I can see that the classification will tell you about the dominant EOFs, but the part about which process, it would come from your interpretation of the results, no?

True, we need to know what each EOF represents to analyze the associated processes, the classification only tells us which EOF is dominant where and helps to objectively find the region with similar values of EOFs and therefore most likely affected by similar processes. We rephrased the sentence to make it clear that knowing dominant processes requires knowledge of what individual EOFs represent.

> L133: Add a reference here to 'silhouette score'. and L146: Reference for soft voting.

We added the references: Rousseeuw (1987) for the silhouette score and Cao et al. (2015) for soft voting.

> Maybe add to the methods section which class number Ks are tested.

We added that information to the end of Sect. 2.2., when describing the silhouette score.

> Fig 3: Did you test if using a higher K value with the lower EOFs, would give a similar result? That is, using K=10 to all the EOFs combination. I understand that the K number was chosen by the silhouette score, but this test could further confirm if your results are dependent on the number of EOFs or on the K number (see Major Comment 1).

Yes, we tested it. Using a high K with low EOF results in the ensemble significantly reducing the number of classes (from 10 to 6-7), but still failing to converge to the same combination of classes when using it multiple times with the same parameters. A more detailed answer is given in response to the major comment 1. We added a short discussion about it into the manuscript as well.

L202: It splits only in 4 classes, because of the K number, not because of the EOF number per se. (see previous comment and Major Comment 1).

Even when setting K=5 or K=6 for the individual GMMs, the ensemble eliminates one (or two) of the classes and the resulting classification looks the same as that obtained with K=4, so it is really a result of the data used for classification, including how much data is included through the selection of the number of EOFs, not just the pre-defined number of classes. We modified the text to make it clearer that it is not the GMM itself that splits the area into four classes (because GMM comes with the predefined K), but the ensemble of GMMs, which can result in a different number of classes than the K selected for the individual GMM. You can see the classification results for one run of the ensemble with 3 EOFs and K=5 in Fig. R3. The model needs one more EOF to find another class.

[Figure]

**Figure R3:** Classification (left) and accompanying likelihood (right) obtained with 3 EOFs by setting the number of classes to five for individual Gaussian Mixture Models. The ensemble of GMMs reduces the number of classes to four and the results are identical to those obtained by setting the number of classes to four (Fig. R1a).

L208-2010: Could this be an indication that you would need one more class to better represent your region? I.e., if you had k=5, then this border might be uniquely classified? (maybe not, because this border remains "difficult" in all other cases). So it might be a hint for an underlying mechanism in this region (for example, see Chafik et al (2023) and Calafat et al (2013))?

It is definitely a "difficult" border to classify, so it is very likely that this is a result of some underlying mechanism in that region. It could be related to the poleward propagation of sea-level fluctuations along the eastern boundary of the North Atlantic, driven by the local winds, as found by Chafik et al. (2023). Mangini et al. (2021; thank you for that reference too) also found a border there associated with the anomalously high sea level. North of it the high sea level is caused by both the northern and the mixed jet cluster, while south of it only by the northern cluster. According to Calafat et al. (2013), the sea level coherence along the Norwegian coast is affected by the variations in the Norwegian Current, which also affects the Atlantic inflow into the North Sea (Winther and Johannessen, 2006), so this border might be affected by that as well. We added a small discussion about this into the manuscript. Even though models with more EOFs can have a lot more classes in the whole area, they never find an additional class there, just shift the border more northward.

> L225: This can also be just because you have too many classes, not necessarily too many EOFs.

This answer is affected by the data set we use for classification. With the new data set there is no such area with lower likelihood in the southern part of the North Sea, which most likely means that it was a result of some processing decisions in the old data set, not related to the number of classes or EOFs. It is not a result of a difference in time spans (the other thing we changed), we checked with the new data set and 1995-2019 time span, and the resulting classification does not have lower likelihood in that region either.

We would, however, still say that the eventual failure of the GMM ensembles to converge is a result of the noise introduced by many EOFs, as well as by the difficulty of measuring distance in high-dimensional spaces, and not the number of classes K, because the best number of classes depends on the input data, including the number of included EOFs. Models with large number of EOFs and small Ks also do not work well, they have low likelihood, if they converge at all. We included a table with the summary of tested models into the revised manuscript, which makes this clearer.

> L229-231: It's not only bathymetry, but the fact that different processes dominate each of those regions. From a sea level perspective: Deeper waters have a significant steric expansion, while that is not present in shallow seas. Shallow seas, in specific the North Sea, is strongly influenced by winds, and that will not happen so much in the open ocean. If you go into a physical oceanographic perspective, then other processes become important.

We apologize for writing it so vaguely, but this is what we meant. Our models definitely cannot see the bathymetry directly, they only detect the changes in sea level variability, which are often controlled by different processes depending on the water depth, and base their classes on that. In the end many of the class borders coincide with the bathymetric features, seen indirectly by the models through the changes in sea level affected by different processes that dominate in the shallow and deep waters. We clarified this in the manuscript and used some of the references you suggested.

> L256: And which one was the recommended number of classes according to the silhouette score? I think it would be good to have these results in the supplementary, so that the reader can see by themselves the difference between a K number that "works better" and one that doesn't.

This answer is based on the results with the new data set. For the Baltic silhouette score recommended only 3 classes (we use 5), for the North 5 (we use 6), and for the Norwegian Sea region 7 (we use 6). The Ks recommended by the silhouette score generally work in the sense that each run of the ensemble produces the same or very similar results, with just a slight shift of one of the borders, but the average likelihood of the whole region is higher for the models we selected. Please note that the Baltic and the Norwegian region include an area on the other side of the Scandinavian Peninsula because of selecting the sub-region as a longitude-latitude box. These areas are included in the models, but always sorted into their own class, which we do not assign a number or discuss. This means that the actual K provided by the silhouette score and given to the model is for these two sub-regions always one higher than what is discussed.

We would prefer to not include the silhouette score for the sub-regions into the manuscript because the selection of classes is not based on them, but we included an explanation of the K selection process. You can however see the silhouette score in Fig. R4, and the results obtained with the Ks recommended by the silhouette score in Fig. R5. In case you are interested in the answer regarding the old data set: the silhouette score recommended 3 classes in the Baltic (real best was 5), only 2 classes in the North Sea (real best was 6), and 10 classes for the Norwegian Sea area (best was actually 8).

a) Baltic Sea

b) North Sea

[Figure]

c) Norwegian Sea

[Figure]

**Figure R4:** Silhouette score for the three sub-regions: Baltic Sea (a), North Sea (b), and Norwegian Sea (c). Differently colored lines represent the silhouette score computed with a different number of EOFs. The number of EOFs used in the manuscript is marked by a thicker line.

> L268-270: Some papers come to mind when reading these lines: Mangini et al (2021) and Hermans et al (2020;2022)

Thank you for the references, they are very useful for explaining some of the classification results and were added to the text, albeit in a different paragraph.

> Figure 5: I didn't fully understand Figure 5, especially columns b to d. It can be ignorance from my side, but I think it's worth adding a bit more explanation to it, since other readers might be confused as well. The first column is clear, as it shows in each row the first 7 EOFs. But the next three columns weren't so clear to me. At each row the classification changes, but the number of EOFs should be the same for the entire column, so what exactly is changing in each row was not clear to me. The way I interpreted it, is that at each row, you added one more EOF to your classification, so the first row had only 1 EOF for the three models (k=10,k-6,k=1), and the second 2 EOFs, and so on... but I'm not sure if that's the correct interpretation.

[Figure]

[Figure]

[Figure]

**Figure R5:** Classification using an ensemble of 200 Gaussian Mixture Models (left) and the respective likelihoods of the model sorting the grid points to that particular class (right) for the Baltic Sea performed using 4 EOFs (a); North Sea using 5 EOFs (b); and part of the Norwegian Sea using 7 EOFs (c) and the number of classes recommended by the silhouette score, which is 3, 5, and 7, respectively. Numbers indicate the assigned classes.

We apologize for not explaining Fig. 5 better. First, we would like to mention that we decided to remove the seventh EOF from the figure because it does not provide any crucial information and excluding it makes the other plots larger and more readable, so Fig. 5 in the revised manuscript only has 6 rows. The classification does not change in every row: Each of the b-d columns show one of the classification models from Fig. 3, and each row shows the class mean for each of the EOFs used for the classification. Apart from assigning classes, GMM also gives the class means and covariance matrices it fits the data to. Since it fits to multivariate Gaussian distributions, for each class it outputs a mean for each EOF used to train it, which is what we show in columns b-d by replacing the values of EOFs at each grid point with mean values from the class assigned to that grid point. Therefore, every plot in column (b) has 10 classes, in (c) 5, and in (d) 4, and the colors show the value of the EOFs. It might sometimes look like the classification changes from row to row because the color is very similar for multiple classes, but that is only because the class means for those particular EOFs and classes are very similar. Since models shown in columns (c) and (d) are based on less than 6 EOFs and they can only provide the class means for the EOFs they are based on, those columns only show the first 5 and 3 EOF maps, respectively. The model shown in column (b) is (now) based on the first 9 EOFs, so it provides class means for them, but we show the first 6 because this is the maximum we could fit onto the figure without reducing the size of individual plot too much. We hope this is clearer. We also included a similar explanation into the manuscript.

> L312: Can you give an example here of a novel idea about the spatial coherence your balance highlighted? (I know you discussed the identified features previously, but quite some of them seemed like you "expected" them…so would be nice to have an example here about a novel spatial structure shown by the GGMs).

It is hard to be absolutely certain that something is completely novel because there is always a chance that we might have missed an article that already found something similar, but for example, our results are showing a different classification than the results by Mangini et al. (2021), possibly because their results are based on jet clusters, i.e., large scale wind only, while ours do not consider driving mechanisms, only sea level itself, which can be driven by other mechanisms as well. We believe our results are best used in combination with a study that uses much longer tide gauge data. For example, Dangendorf et al. (2014) found differences in sea level drivers between the western and northern North Sea and southern and eastern based on tide gauge series. Our results show that their western coast findings based on British tide gauges are most likely valid for the majority of the North Sea, while those on the east coast generally apply only in a narrow stretch near the coasts. We added these and some more examples into the manuscript, together with a discussion of the expected results that match previous studies.

> L321-323: And what is the significance of this "spread"? More variability in those classes?

Yes, exactly. We added that to the text.

> L351-358: This is just my opinion, so not a "requirement" as a reviewer. This entire paragraph is describing characteristics of spatial pattern classification methods in general. Most of it would also be true for SOM and K-means, for example. And it doesn't seem to be the main take-away message of your article, but just characteristics of GGM. I would suggest ending with a stronger message about your study in specific.

Thank you for your opinion. We kept this paragraph, as it is valid for GMMs and it explains in short the benefits of the classification, but we also added a paragraph focused on the benefits of GMMs and on our results in particular to conclude the paper.

**Technical/editorial comments**

L18: "so" – suggest changing it for "thus" or "therefore", to avoid repetition (L16), and less colloquial also.

L38: I'm not sure if you can/should start a sentence with "therefore".

L80-86: you repeat "While" three times in these lines. Suggest to modify a bit to avoid repetition.

L158: Add a comma after voting.

L246: Referring here to Figure 5a was a bit bothersome for me, and I'm not sure if it's necessary. I went down to check it, and then got a bit lost in the text.

L334: Suggest adding "(classes)" after "mixture components".

Section 4: This is a "summary" not a "conclusion".

Thank you for your comments. We made the necessary modifications.

**References by reviewer**

Calafat, F. M., Chambers, D. P., and Tsimplis, M. N.(2013), Inter-annual to decadal sea-level variability in the coastal zones of the Norwegian and Siberian Seas: The role of atmospheric forcing, Geophys. Res. Oceans, 118, 1287–1301, doi:10.1002/jgrc.20106.

Camargo, C. M. L., Riva, R. E. M., Hermans, T. H. J., Schütt, E. M., Marcos, M., Hernandez-Carrasco, I., and Slangen, A. B. A.: Regionalizing the sea-level budget with machine learning techniques, Ocean Sci., 19, 17–41, https://doi.org/10.5194/os-19-17-2023, 2023.

Chafik, L., Nilsson, J., Rossby, T., & Kondetharayil Soman, A.(2023). The Faroe-Shetland Channel Jet: Structure, variability, and driving mechanisms. Journal of Geophysical Research: Oceans, 128, e2022JC019083. https://doi.org/10.1029/2022JC019083

Frederikse, T. and Gerkema, T.: Multi-decadal variability in seasonal mean sea level along the North Sea coast, Ocean Sci., 14, 1491–1501, https://doi.org/10.5194/os-14-1491-2018, 2018.

Hermans, T. H. J., C. A. Katsman, C. M. L. Camargo, G. G. Garner, R. E. Kopp, and A. B. A. Slangen, 2022: The Effect of Wind Stress on Seasonal Sea-Level Change on the Northwestern European Shelf. Climate, 35, 1745–1759, https://doi.org/10.1175/JCLI-D-21-0636.1.

Hermans, T. H. J., Le Bars, D., Katsman, C. A., Camargo, C. M. L., Gerkema, T., Calafat, F. M., et al. (2020). Drivers of interannual sea level variability on the northwestern European shelf. Journal of Geophysical Research: Oceans, 125, e2020JC016325. https://doi.org/10.1029/2020JC016325

Mangini, F., Chafik, L., Madonna, E., Li, C., Bertino, L. and Nilsen, J.E.Ø., 2021. The relationship between the eddy-driven jet stream and northern European sea level variability. Tellus A: Dynamic Meteorology and Oceanography, 73(1), p.1886419.DOI: https://doi.org/10.1080/16000870.2021.1886419

Thompson, P. R. and Merrifield, M. A.: A unique asymmetry in the pattern of recent sea level change, Geophys. Res. Lett., 41, 7675–7683, https://doi.org/10.1002/2014GL061263, 2014.

**Additional references**

Cao, J., Kwong, S., Wang, R., Li, X., Li, K. and Kong, X. (2015): Class-specific soft voting based multiple extreme learning machines ensemble, Neurocomputing, 149(A), 275-284, https://doi.org/10.1016/j.neucom.2014.02.072

Rousseeuw, P. J. (1987): Silhouettes: A graphical aid to the interpretation and validation of cluster analysis, Journal of Computational and Applied Mathematics, 20, 53-65, https://doi.org/10.1016/0377-0427(87)90125-7.

Winther, N. G. and Johannessen, J. A. (2006): North Sea circulation: Atlantic inflow and its destination. Journal of Geophysical Research, 111(C12018), https://doi.org/10.1029/2005JC003310.

---

## Author Comment (AC2)

**Response to Reviewer#2**

Dear authors, congratulations for the performed work.

The manuscript that you presented includes a really interesting and robust technique (GMM) that was used for oceanic regionalisation considering SSH and presenting accurate results. Additionally, I find this technique really promising because it can be applied considering other variables (SST, currents, SSS, chlorophyll concentration, turbidity, ...) from different databases (in situ, remote sensing, numerical models).

The manuscript is really well writen, in good English, easy to follow and to understand. The figures are clear and necessary, the conclusions are in the line of the obtained results and the references are up-to-date.

I have some comments that I expect could help to improve the manuscript.

We thank the reviewer for their supportive review and extremely helpful questions and comments. Their contribution was crucial in the improvement of the manuscript and is acknowledged in it (see acknowledgements). In the following text we answer each of the comments individually.

**Major comment**

The mayor comment that I have is that I missed the explanation of why these regions were split (Figures 3 and 4) and if they correspond with bathymetry/hydrodynamic characteristics. The technique is really good and the results are promising, but I miss here a little about the physics of the regions, justifying why the classification method selected those regions and which specificities each one of them has that differs from the others, reinforcing and validating the obtained results.

Thank you for your recommendation. We originally tried to somewhat refrain from it because GMM is not a physical model and based on purely its results, without using other methods, we cannot explain the physical processes; besides, the physical drivers of sea level variability are the topic of another manuscript of ours, currently under review. But you are right that the manuscript lacks ocean science, so we added more discussions for some of the regions and dominant processes in them, based primarily on other people's works. See also our responses to similar comments made throughout the manuscript by Reviewer#1.

**Other comments**

Line 80: I would like to ask the authors why they selected the period 1995 - 2019. The authors explained why they start in 1995, but not why they end in 2019. The selected database is now available until August 2022. I agree to have entire years, so to not considered 2022. But, why the authors did not considered 2020 and 2021?

The honest answer is: Because we started this work a long time ago, when only data until mid-2020 was available. But since now not only newer data is available, but the processing chain for the data set had been modified since we originally downloaded the data set (see description of the changes to the processing chain at https://catalogue.marine.copernicus.eu/documents/QUID/CMEMS-SL-QUID-008-032-068.pdf) we decided to re-do the experiments with the new expanded and improved data set. This has caused some changes in the classification results (e.g., needing a different number of EOFs to achieve the same number of classes or losing some of the classes), stemming from both the processing changes and from the change in the time span, visible by comparing the results with the old 1995-2019, new 1995-2019, and the new 1995-2021 data sets. You can see the new results

(replacement for Figs. 3 and 4 in the manuscript) in Figs. R1 and R2 and compare them with the results obtained with the 1995-2019 time period with the new data set on Fig. R3 (intermediate model only), as well as with the results with the old data set from the manuscript.

---

## Author Response (AR1)

**Final Author Reply to the Editor**

Dear Dr. Alvera-Azcárate,

Thank you for giving us the opportunity to submit the revised draft of our manuscript titled "**Unsupervised classification of the northwestern European seas based on satellite altimetry data**" to the *Special Issue for the 54th International Liège Colloquium on Machine Learning and Data Analysis in Oceanography* in *Ocean Science*. We appreciate the time and effort that you and the reviewers have dedicated to providing your valuable feedback on our manuscript. We are grateful to the reviewers for their insightful comments on our paper and have incorporated changes to reflect their suggestions. We acknowledge the contribution of the reviewers in the manuscript. Because one comment by Reviewer#2 lead us to download a new data set and re-do all our experiments with it, which affects the replies to some of the other comments by both reviewers, we reply to this particular comment first to explain the changes we made and point out all the differences in the results and the manuscript this has caused. After that we reply point-by-point to all the other comments and concerns by both reviewers. All excerpts from the manuscript are denoted by quotation marks, with changes highlighted in blue. We have also highlighted the changes within the manuscript.

**Summary of the changes in the classification results**

**Comment by Referee#2**

Line 80: I would like to ask the authors why they selected the period 1995 - 2019. The authors explained why they start in 1995, but not why they end in 2019. The selected database is now available until August 2022. I agree to have entire years, so to not considered 2022. But, why the authors did not considered 2020 and 2021?

The honest answer is: Because we started this work a long time ago, when only data until mid-2020 was available. But since now not only newer data is available, but the processing chain for the data set had been modified since we originally downloaded the data set (see description of the changes to the processing chain at https://catalogue.marine.copernicus.eu/documents/QUID/CMEMS-SL-QUID-008-032-068.pdf) we decided to re-do the experiments with the new expanded and improved data set. This has caused some changes in the classification results (e.g., needing a different number of EOFs to achieve the same number of classes or losing some of the classes), stemming from both the processing changes and from the change in the time span, visible by comparing the results with the old 1995-2019, new 1995-2019, and the new 1995-2021 data sets. We no longer discuss the former data set and assume that the newer data set is improved and thus classification results obtained with it are more realistic. The differences due to different time spans might have physical relevance and show that perhaps this method can be used to track the changes in large scale sea level patterns, but we do not focus on that in the current manuscript.

With the new data set, when considering the whole northwestern European continental shelf (Fig. 3) the simplest model remains the same, the intermediate model has one fewer class (class O1 from the original manuscript is not visible in this model), and the complex model remains virtually the same. Class O1 is also found by this model, suggesting that the signals responsible for it are still present in the data, but now only in the higher EOFs, which is why a model using only five of them did not capture it. The area with low likelihood in the southern North Sea visible on Figure 3c in the original manuscript is also gone, suggesting that it was a result of data processing in the old data set. Additionally, with this data set, we obtain almost identical K=10 classification with 8, 9, 10, and 11 EOFs, of which we decided to display the results with 9 EOFs because it is the lowest number of

EOFs (meaning the fastest model) for which the model reaches the minimal likelihood of 0.95 averaged over the whole region.

Of the sub-regions (Fig. 4), the Baltic Sea region classification remains the same and the classification of the North Sea has a difference only along the southern coast, most likely for the same reason as the difference in likelihood in the whole area model, but we now need 5 EOFs to achieve that instead of only 3. This unfortunately means that it is no longer possible to directly show this model in the abstract EOF space because it now requires 5 dimensions. Fig. 6 therefore now has only the simple model of the whole region and the discussion of the classes in the abstract space is shortened. For the Norwegian Sea we did not manage to obtain the same number of classes as before, the highest K with stable results is 6. The likelihood for this model is also lower than previously.

We replaced the Figs. 3, 4, 5, and 6 and modified the text in Sect. 2.1 (data set and processing description), 2.3 (influence of the different data set to the selection of the ensemble hyperparameters), 3 (results), and 4 (summary and conclusions) to reflect these changes.

**Section 2.1**

[revised manuscript text omitted]
 main difference here is that the text pertaining to the North Sea model is removed.)

**Section 4**

"The largest number of classes with which we can achieve robust results for our region is 10, and we need to use between 8 and 11 EOF maps as input."

**Response to Referee#1**

Review of "Unsupervised classification of the Northwestern European seas based on satellite altimetry data" by Poropat et al., 2023

In this work, Poropat and colleagues use a Gaussian Mixture Model (GMM) to identify coherent regions of sea-level variability in the Northwestern European Seas. They show how the number of EOFs and the number of classes (mixtures) in the models are important parameters that can result in different patterns, but the main classification remains the same, showing the robustness of their method. The work is focused on the method itself, and I personally missed a bit more discussion into what the identified patterns could actually mean. Nonetheless, I believe this is an important work and a good addition to the scientific community, which could be the base for more process-based studies in the future.

We thank the reviewer for their positive evaluation and valuable questions and comments that helped us to improve our manuscript. We added more discussion about the physical processes that might be related to the identified patterns. We address all the comments in the following text.

**Major comments**

Number of EOFs X Number of Classes: The results from Section 3.1 are very interesting. But for me it wasn't clear if the different classifications are from adding more EOFs or from changing the number of classes. The authors used a "non subjective" way to choose the number of classes, which is important for several reasons.

But I do wander, if the results from Figure 3 would be the similar if they fixed the number of EOFs, and just changed the number of classes. Was this tested? Because during the entire Section 3.1, the authors presents the results as an effect of adding more EOFs, but it could just be due to adding more classes. So testing the classification for fixed number of EOFs and changing the number of classes would make their results and discussion more robust.

We apologize this was not clearly explained in the paper. We tested all class numbers from 2 to 11 for each of the presented numbers of EOFs (and for many not presented). The silhouette score is a very useful metric that often gives a good recommendation for the optimal number of classes, but it does not always work perfectly, so we tested all the combinations of number of EOFs and number of classes to confirm whether its recommendation was correct. In the end we picked the number of EOFs somewhat arbitrarily to show different levels of complexity and used the number of classes that work best for each given number of EOFs based on the criteria presented in Sect. 2.3., which in case of the whole northwestern European shelf coincides with the recommendation of the silhouette score. In case of the sub-regions it does not, but the discussion about that is in the answer to one of the minor comments later in the text. We added the following explanation to the end of Sect. 2.2., when describing the silhouette score: "However, since the silhouette score does not always work perfectly, we test all the class numbers between 2 and 11 for each number of EOFs and our tests

confirm that the silhouette score in our case indeed recommends the best option. The summary of the tests is given in Table A1.", as well as to the beginning of Sect. 3.1: "Testing the models confirms that the number of classes recommended by the silhouette score is correct, as can be seen from Table A1 in the appendix. Typically, when using high number of EOFs with a smaller number of classes or vice versa, the models either do not work at all, i.e., re-running the ensembles results in a different classification, or work, but have lower likelihood than the models we selected. In some cases when the difference between the number of classes $K$ given to the individual GMMs and the optimal number of classes is small, the ensemble is able to find the optimal number of classes on its own."

> Literature & Discussion: I missed the "discussion" section, but I don't think it's reasonable to ask the authors to add an entire discussion section, just maybe in some locations when describing the features identified, might be a good addition to refer to some papers that could bring some insights into the processes behind these patterns. For example, regarding the features with the classification, there are some works that could highlight some of the processes behind the identified patterns (e.g., Mangini et al, 2021; Hermans et al, 2020; Frederikse et al., 2018, Chafik et al (2023), Calafat et al (2013), among others). Also, I would expect the authors to acknowledge the works of Thompson & Merrifield (2014) and of Camargo et al (2023). Both works have performed classification of ocean regions based on sea level data, and seem relevant for the present work. The ocean regions from Thompson & Merrifield have been widely used in sea level studies. The work of Camargo et al (2023) used two classifying methods to identify coherent regions of sea level variability. One of the methods of Camargo et al (2023) was SOM, which Poropat et al mention on the introduction, and hence acknowledging this work there seems fitting.

We thank the reviewer for the comment and the references, they were very useful for improving the discussion of our results. We believe that an in-depth analysis of the mechanisms contributing to sea level variability requires applying additional methods and deserves a separate paper. GMMs on their own can only find patterns, not the causes of those patterns, so here we would like to focus on finding regions of coherent sea level variability and in our next manuscript (currently under review) we focus on the driving processes of sea level in each of the regions we found here. We realize, however, that the manuscript should nonetheless include at least some discussion of the physical background and the mechanisms driving sea level, so we included more analysis of our results, trying to explain the dominant processes in at least some of the classes/regions and connect our findings with those from other works, including the references listed by the reviewer. We here note only some changes related to the addition of the discussion of the results and references to previous works because many others are related to one of the reviewer's other, more specific, questions, so to avoid repetition we included them there, or are related to the different results obtained with the new data set and are therefore already listed in the Summary of the changes due to the new data set:

"There have also been a few attempts to apply more complex classification or clustering methods to sea level, e.g. Scotto et al. (2010) used agglomerative hierarchical methods to group time series in the North Atlantic Ocean based on their posterior predictive distributions for extreme values, Thompson and Merrifield (2014) applied it to the whole ocean, while Barbosa et al. (2016) used wavelet-based clustering to find regions with similar sea level records in the Baltic Sea."

"Camargo et al. (2023) applied SOM, as well as a network detection approach (δ-MAPS) to regionalize the world's sea level budget."

"In the simplest model (Fig. 5d and Fig. 3a) the GMMs capture only the rough patterns of the first three EOF maps, which mostly represent the sea level rise (EOF1) and the North Atlantic Oscillation (EOF2 and EOF3). All three EOFs contain additional processes but those are not as easily identifiable. The border between the Baltic (class B) and the North Sea (class N) is visible on most EOF maps (column a), except the second one, showing that since the Baltic is an enclosed sea,

almost all processes in it differ from those in the neighboring North Sea at least to some extent. The border based on the continental shelf break can also be seen on most EOFs, but is most visible in EOF 2 and 3, demonstrating how clearly the large difference in ocean depth affects sea level. The only border in the simplest model (column d) that is not based on bathymetry, i.e. the border between the North Sea and the rest of the continental shelf, just south of 60 °N, is determined only by the not very steep gradients in the first three EOF maps in that location, which is probably why the individual GMMs do not completely agree where to place it, resulting in lower likelihood around the border."

**Minor comments**

> L47-50: Isn't this true for other classification/clustering methods also? Once clusters are identified, it can be transformed in a mask to isolate regions…

Yes, it is true. We modified the sentence to make it clear that this is a possible use of all clustering methods, not specific to GMM: "Because clustering methods such as GMM give a class for every data point, the results from it not only provide an insight into the patterns of sea level variability, but can also be used as a mask to isolate a region and focus on the dominant processes in it without being affected by the noise from everything in the neighboring areas."

> L93-94: It wasn't clear for me if it's common to use EOFs as input for GGMs, or if this was a "novel" approach that the authors found to reduce noise? Would be good to know in both cases.

It was used by all the studies we cited that were using GMMs with temperature and salinity profiles, so it is a common approach with GMMs. It is "novel" in regard to classification based on sea level; the studies we found that applied other clustering methods to sea level data usually considered whole time series. We added a sentence to make this point clearer: "This is a standard procedure when applying GMM to other data sets (e.g., Maze et al., 2017; Thomas et al., 2021), but is rather uncommon when using other clustering techniques on sea level data, where previous studies typically used the whole time series (e.g., Liu and Weisberg, 2005; Liu et al., 2008)."

> L101-102: Just a comment, but this is also true for SOM.

True, they are similar methods in that regard.

> L124: How would the mean values give information about processes associated? I can see that the classification will tell you about the dominant EOFs, but the part about which process, it would come from your interpretation of the results, no?

True, we need to know what each EOF represents to analyze the associated processes, the classification only tells us which EOF is dominant where and helps to objectively find the region with similar values of EOFs and therefore most likely affected by similar processes. We rephrased the sentence to make it clear that knowing dominant processes requires knowledge of what individual EOFs represent: "The mean values which define each class in our case give us information about which EOFs and, consequently, if we are able to identify which processes a particular EOF represents, which processes are dominant in that region."

> L133: Add a reference here to 'silhouette score'. and L146: Reference for soft voting.

We added the references: Rousseeuw (1987) for the silhouette score and Cao et al. (2015) for soft voting.

> Maybe add to the methods section which class number Ks are tested.

We added that information to the end of Sect. 2.2., when describing the silhouette score. The exact change in the manuscript is already given as a response to the reviewer's first major comment.

> Fig 3: Did you test if using a higher K value with the lower EOFs, would give a similar result? That is, using K=10 to all the EOFs combination. I understand that the K number was chosen by the silhouette score, but this test could further confirm if your results are dependent on the number of EOFs or on the K number (see Major Comment 1).

Yes, we tested it. Using a high K with low EOF results in the ensemble significantly reducing the number of classes (from 10 to 6-7), but still failing to converge to the same combination of classes when using it multiple times with the same parameters. A more detailed answer is given in response to the major comment 1, as is the excerpt from the manuscript.

> L202: It splits only in 4 classes, because of the K number, not because of the EOF number per se. (see previous comment and Major Comment 1).

Even when setting K=5 or K=6 for the individual GMMs, the ensemble eliminates one (or two) of the classes and the resulting classification looks the same as that obtained with K=4, so it is really a result of the data used for classification, including how much data is included through the selection of the number of EOFs, not just the pre-defined number of classes. We modified the text to make it clearer that it is not the GMM itself that splits the area into four classes (because GMM comes with the predefined K), but the ensemble of GMMs, which can result in a different number of classes than the K selected for the individual GMM: "Despite having one border which is harder to define for individual GMMs, the ensemble classification is extremely robust; increasing the number of classes in the individual ensembles to five or even six results in the exact same classification because the ensemble removes the unnecessary classes. This shows that based on the largest processes contained in the first three EOFs, there are exactly four regions with distinct sea level patterns in the northwestern European seas."

> L208-2010: Could this be an indication that you would need one more class to better represent your region? I.e., if you had k=5, then this border might be uniquely classified? (maybe not, because this border remains "difficult" in all other cases). So it might be a hint for an underlying mechanism in this region (for example, see Chafik et al (2023) and Calafat et al (2013))?

It is definitely a "difficult" border to classify, so it is very likely that this is a result of some underlying mechanism in that region. We added a small discussion about this into the manuscript: "It is most likely related to some of the underlying mechanisms in that region, such as the poleward propagation of sea level fluctuations along the eastern boundary of the North Atlantic, as found by Chafik et al. (2023) or the variations in the Atlantic inflow into the North Sea (Winther and Johannessen, 2006). North Sea sea level is also highly affected by wind and atmospheric pressure, and which of them dominates depends on the location (Dangendorf et al., 2014). The border corresponds well with the border found by Mangini et al. (2021) between the dominant influence of different jet clusters which represent large-scale atmospheric circulation patterns."

> L225: This can also be just because you have too many classes, not necessarily too many EOFs.

This answer is affected by the data set we use for classification. With the new data set there is no such area with lower likelihood in the southern part of the North Sea, which most likely means that it was a result of some processing decisions in the old data set, not related to the number of classes or EOFs. It is not a result of a difference in time spans (the other thing we changed), we checked with the new data set and 1995-2019 time span, and the resulting classification does not have lower likelihood in that region either. We would, however, still say that the eventual failure of the GMM ensembles to converge is a result of the noise introduced by many EOFs, as well as by the difficulty of measuring distance in high-dimensional spaces, and not the number of classes K, because the best number of classes depends on the input data, including the number of included EOFs. Models with large number of EOFs and small Ks also do not work well, they have low likelihood, if they

converge at all. We included a table with the summary of tested models into the revised manuscript (Appendix A), which makes this clearer.

> L229-231: It's not only bathymetry, but the fact that different processes dominate each of those regions. From a sea level perspective: Deeper waters have a significant steric expansion, while that is not present in shallow seas. Shallow seas, in specific the North Sea, is strongly influenced by winds, and that will not happen so much in the open ocean. If you go into a physical oceanographic perspective, then other processes become important.

We apologize for writing it so vaguely, but this is what we meant. We clarified this in the manuscript and used some of the references you suggested: "Note that despite these class borders coincide perfectly with the steep changes in the ocean depth or with the coastlines, the GMMs do not explicitly include those things; the classification is based solely on the differences in sea level variability caused by different dominant processes on the continental shelf and in deep waters, as well as by the coastlines directing the circulation in the enclosed seas. The steric contribution is known to be prevalent in the deep ocean, while in coastal regions complex bathymetry, local circulation, and forcing from the atmosphere and rivers can be more significant (e.g., Passaro et al., 2015)."

> L256: And which one was the recommended number of classes according to the silhouette score? I think it would be good to have these results in the supplementary, so that the reader can see by themselves the difference between a K number that "works better" and one that doesn't.

This answer is based on the results with the new data set. For the Baltic silhouette score recommended only 3 classes (we use 5), for the North 5 (we use 6), and for the Norwegian Sea region 7 (we use 6). The Ks recommended by the silhouette score generally work in the sense that each run of the ensemble produces the same or very similar results, with just a slight shift of one of the borders, but the average likelihood of the whole region is higher for the models we selected. Please note that the Baltic and the Norwegian region include an area on the other side of the Scandinavian Peninsula because of selecting the sub-region as a longitude-latitude box. These areas are included in the models, but always sorted into their own class, which we do not assign a number or discuss. This means that the actual K provided by the silhouette score and given to the model is for these two sub-regions always one higher than what is discussed.

We would prefer to not include the silhouette score for the sub-regions into the manuscript because the selection of classes is not based on them, but we included an explanation of the K selection process: "We select the models based on the principles presented in Sect. 2.3. We again calculate the silhouette score for all combinations of the number of EOFs and class numbers in all three sub-regions, but there are better class numbers than those recommended by the silhouette score (3, 5, and 7 classes). In the Baltic and the Norwegian Sea models the likelihood of the models we present is significantly higher than the likelihood of the model recommended by the silhouette score, while for the North Sea they are equally good, but the model we present has a higher number of classes, thus allowing a more detailed subdivision. This demonstrates that while the silhouette score is a good tool to give an estimate of the number of mixture components, it does not always give the best result. One should always try the model with several options to find the best solution to their specific classification problem."

> L268-270: Some papers come to mind when reading these lines: Mangini et al (2021) and Hermans et al (2020;2022)

Thank you for the references, they are very useful for explaining some of the classification results and were added to the text, albeit in a different paragraph.

> Figure 5: I didn't fully understand Figure 5, especially columns b to d. It can be ignorance from my side, but I think it's worth adding a bit more explanation to it, since other readers might be confused as well. The first column is clear, as it shows in each row the first 7 EOFs. But the next three columns weren't so clear to me. At each row the classification changes, but the number of EOFs should be the same for the entire column, so what exactly is changing in each row was not clear to me. The way I interpreted it, is that at each row, you added one more EOF to your classification, so the first row had only 1 EOF for the three models (k=10,k-6,k=1), and the second 2 EOFs, and so on… but I'm not sure if that's the correct interpretation.

We apologize for not explaining Fig. 5 better. First, we would like to mention that we decided to remove the seventh EOF from the figure because it does not provide any crucial information and excluding it makes the other plots larger and more readable, so Fig. 5 in the revised manuscript only has 6 rows. We added the following explanation of Fig. 5: "Apart from assigning classes, GMM also gives the class means and covariance matrices it fits the data to, which in our case is a class mean for each EOF used to train it. Therefore, to see how the models from Fig. 3 determine the class borders, we can compare the EOF maps (Fig. 5a) with the maps in which we replace the values of EOFs at each grid point with mean values from the class assigned to that point (Fig. 5b, c, and d). This can reveal two things: 1) a comparison of the class mean EOFs with the original EOFs indicates how well the model fits to the data; and 2) the difference in class means between two classes can tell us which EOFs are responsible for that class border. The accompanying principal component time series can be seen in Fig. B1 in the appendix."

> L312: Can you give an example here of a novel idea about the spatial coherence your balance highlighted? (I know you discussed the identified features previously, but quite some of them seemed like you "expected" them…so would be nice to have an example here about a novel spatial structure shown by the GGMs).

We added the following: "Unlike the simplest and even the intermediate model, in which the class borders mainly coincide with bathymetry and could be determined without any particular method by selecting depth ranges or, in case of the Baltic Sea, based on the coastline, the most complex model separates the ocean into regions that are not so obvious. The difference between the east, west and south coast in the North Sea is also found by Dangendorf et al. (2014) and Frederikse and Gerkema (2018) based on tide gauges, but our work shows that their findings for the British coast are most likely also valid for most of the North Sea, while those for the Norwegian coast apply only on a narrow stretch along the coast. Furthermore, while different drivers of sea level variability in the North Sea have been already studied by many (e.g., Dangendorf et al., 2014; Frederikse and Gerkema, 2018; Hermans et al., 2020), the other shelf areas and especially of the adjoining deep ocean basins are less well understood, so our results can help us determine which regions should be studied separately."

> L321-323: And what is the significance of this "spread"? More variability in those classes?

Yes, exactly. We added the following to the text to hopefully better explain Fig. 6: "The Baltic class (B), being very uniform in regards to sea level variability, has a narrow range in the EOF space. North Sea class (N) is also quite uniform in the second and third EOF, but has a wide range in the first one because there is a large spatial gradient in the sea level trend there. The coastal class (C) is the most widespread of them, covering different levels of variability in the first three EOFs along the whole coastline. Finally, the open ocean class (O) covers the largest area in space, but since it has less variability than the coastal class in the first three EOFs, it is less widespread in the EOF space. Even though the border between classes C and O is very compact in the EOF space, it is also apparently easily seen by the model and, when transferred to a map, almost perfectly follows the continental shelf break. This shows that, while some changes in variability might be small at the shelf break, they are very clearly defined."

L351-358: This is just my opinion, so not a "requirement" as a reviewer. This entire paragraph is describing characteristics of spatial pattern classification methods in general. Most of it would also be true for SOM and K-means, for example. And it doesn't seem to be the main take-away message of your article, but just characteristics of GGM. I would suggest ending with a stronger message about your study in specific.

Thank you for your opinion. We kept this paragraph, as it is valid for GMMs and it explains in short the benefits of the classification, but we also added a paragraph focused on the benefits of GMMs and on our results in particular to conclude the paper: "We find that western Baltic has significantly different variability from the rest of the Baltic Sea and should be considered separately, as well as how to separate the North Sea from the rest of the continental shelf. We confirm previous findings that the North Sea has differences between eastern, southern, and western coast because of the different atmospheric drivers along those coasts, but also show that the sea level observed on the British coast is representative for the whole North Sea, while that observed on the eastern coast is more localized. We find that the sea level in the Barents Sea considerably differs from that along the rest of the continental shelf break but only after including higher order EOFs. We can use these results to further study these regions and determine what is the cause of different sea level changes in them."

**Technical/editorial comments**

L18: "so" – suggest changing it for "thus" or "therefore", to avoid repetition (L16), and less colloquial also.

L38: I'm not sure if you can/should start a sentence with "therefore".

L80-86: you repeat "While" three times in these lines. Suggest to modify a bit to avoid repetition.

L158: Add a comma after voting.

L246: Referring here to Figure 5a was a bit bothersome for me, and I'm not sure if it's necessary. I went down to check it, and then got a bit lost in the text.

L334: Suggest adding "(classes)" after "mixture components".

Section 4: This is a "summary" not a "conclusion".

We thank the reviewer for pointing these out. We checked and "Therefore" can be used at the beginning of the sentence, so we kept it. We made all the other suggested modifications.

**Response to Referee#2**

Dear authors, congratulations for the performed work.

The manuscript that you presented includes a really interesting and robust technique (GMM) that was used for oceanic regionalisation considering SSH and presenting accurate results. Additionally, I find this technique really promising because it can be applied considering other variables (SST, currents, SSS, chlorophyll concentration, turbidity, ...) from different databases (in situ, remote sensing, numerical models).

The manuscript is really well writen, in good English, easy to follow and to understand. The figures are clear and necessary, the conclusions are in the line of the obtained results and the references are up-to-date.

I have some comments that I expect could help to improve the manuscript.

We thank the reviewer for their supportive review and extremely helpful questions and comments.

**Major comment**

> The mayor comment that I have is that I missed the explanation of why these regions were split (Figures 3 and 4) and if they correspond with bathymetry/hydrodynamic characteristics. The technique is really good and the results are promising, but I miss here a little about the physics of the regions, justifying why the classification method selected those regions and which specificities each one of them has that differs from the others, reinforcing and validating the obtained results.

We thank the reviewer for their recommendation. We originally tried to somewhat refrain from it because GMM is not a physical model and based on purely its results, without using other methods, we cannot explain the physical processes; besides, the physical drivers of sea level variability are the topic of another manuscript of ours, currently under review. But it is true that the manuscript lacks ocean science, so we added more discussions for some of the regions and dominant processes in them, based primarily on other people's works. You can see the changes we made in the manuscript in our response to the second major comment by Reviewer#1.

**Other comments**

> Line 80: I would like to ask the authors why they selected the period 1995 - 2019. The authors explained why they start in 1995, but not why they end in 2019. The selected database is now available until August 2022. I agree to have entire years, so to not considered 2022. But, why the authors did not considered 2020 and 2021?

The response to this question is at the beginning, as Summary of the changes in the classification results.

> Line 84: The authors mentioned that "We also remove the seasonal cycle by subtracting the climatology calculated from the 25 years of data in order to focus on the non-seasonal variability.". And what about the trend? It is maintained? If yes, it could be possible that the existence of different trends in the study regions affect to the classification?

The trend is maintained, the spatial patterns of the sea level trend are included in the first EOF map. So yes, the existence of different trends in the study area definitely affects the classification. We added a sentence to the manuscript to explicitly state that: "The data set still contains the trend, i.e., the sea level rise signal and its spatial patterns." and later, in Sect. 3.3, noted that the trend is included in EOF 1 when discussing the contribution of each EOF to the classification.

> Line 133: For the Silhouette coefficient I recommend to include a reference. Here I suggested one, but it could be another one. Filaire, T., 2018. Clustering on mixed type data, a proposed approach using R. https://towardsdatascience.com/clustering-on-mixed-type-data-8bbd 0a2569c3

Thank you for the suggested reference. Unfortunately, none of the authors have access to this article, but we found another reference we can use instead: Rousseeuw (1987).

> Line 136: What is the mean S? I don't understand very well how the S score was used. On my understanding, to define the number of components (K), normally S is iterated several times starting from K=2 to higher values, and then, the K that gives the best S value is selected. Please, if possible, include here a deep explanation.

We modified the explanation of the silhouette score in the manuscript to make it clearer: "Silhouette score for each sample (grid point) is computed as: [equation (5)], where $a$ is the mean intra-cluster distance between the sample $i$ and all other samples from the same cluster and $b$ is the mean

nearest-cluster distance between sample *i* and all samples from the nearest cluster. To determine the best number of classes, we use the *S* averaged over all samples."

> Line 193: This part is not really clear for me. I understand that 11 EOFs represented the 85% of the variability. But why considering 11 EOFs when the Silhouette coefficient for this model present the lowest values of all the run models (Figure 2)?

We only used the silhouette score to find the best number of classes for a specific number of EOFs, not for comparisons between different EOFs. One of the effects of the curse of dimensionality is that as we increase the number of dimensions (EOFs) the distance between any two data points becomes more similar and less meaningful, which also affects the silhouette score. In the end we used the model which allows the separation into the highest number of classes because we wanted to see the classes which would otherwise be not so obvious like the classes obtained with 3 EOFs are. We added a sentence explaining that into Sect. 3.1 when describing the selection of K with the silhouette score: "As we increase the number of dimensions (EOFs), the distance between any two points becomes more similar and less meaningful, which also lowers the silhouette score for all class numbers for a given high number of EOFs, so it does not make sense to compare the silhouette score for different number of EOFs."

> Line 223: "The likelihood for the classification in the southern part of the North Sea is also significantly reduced, suggesting that the models struggle to properly classify this region, possibly because this many principal components introduce a lot of noise.". And it could not be related with the value of the S score that is the lowest of all the run models?

This is a very good question. Originally the answer would have been yes, both the low silhouette score and the low likelihood in that region might be a result of the difficulties of considering distance in a 11-dimensional space, but after changing the data set that area of low likelihood is gone in both 1995-2019 and 1995-2021 results, so it was most likely a result of something related to data processing. This part is no longer in the manuscript.

> Figure 4: The manuscript did not include an explanation of why those K were selected. Additionally, the S score is not presented. I recommend to add this information.

> Line 255: "Note that this number of classes is not chosen with the silhouette score.". If the S score is not used in this region, why the authors chose 4 classes?

We tested all the Ks around those determined by the silhouette score and chose to use and present the results which fulfilled the requirements for the "best" model we listed in Sect. 2.3. Since in the end the silhouette score is not used for the number of classes in any of the subregions, we believe a figure with it would not fit well into the article. We added the explanation of how we select K into the manuscript, the exact text of which can be seen as a response to one of Reviewer#1's minor comments.

> Figure 5 is somewhat confusing. Can you please add more explanations to be fully understandable?

Yes, we apologize, we realize that the explanation we provided for this figure was lacking and easily misunderstood, so we added a better one into the revised manuscript. For the exact modification, see the reply to a similar comment by Reviewer#1.

**Additional changes to the manuscript**

We also made the following changes unrelated to the comments by the reviewers:

- We added a secondary affiliation, changed the contact information and added another funding source for the first author due to moving to a new institution.

- We realized that we never explicitly mentioned the split between the training and the test set in the text, so we added that to Sect. 2.1: "We train all our models on 90 % randomly selected grid points and use the remaining 10 % as a test set, to ensure that the model is not only fitted to the training set but is able to generalize to the data points that were not used for training." Simultaneously, we removed the dots representing the exact location of the test set grid points in Figs. 3 and 4 because that is irrelevant and does not contribute to the analysis, while unnecessarily cluttering the figures.
- We added the following sentences to Sect. 2.3: "Despite sometimes resulting in different classifications, the probability given by the GMM is almost always very close to one inside the class and lower only along the class borders, making it hard to assess which classification is better." and "This is a big advantage compared to using only individual GMMs because it makes it easier to see how stable each classification is." to better emphasize the advantages of an ensemble compared to an individual GMM.
- In Sect. 2.3 we added the following: "Please note that matching classes based on correlation requires at least three points; using the ensemble in this manner does not work for only one or two EOFs." because we forgot to state it before, and it is important.
- We reduced the number of EOFs (rows) shown in Fig. 5 from 7 to 6 to make it more readable.
- We fixed some grammatical and spelling errors we found.
- We changed the color schemes in our figures to make them more suitable for readers with color vision deficiencies.